# Discovering Model Structure of Dynamical Systems with Combinatorial Bayesian Optimization

**Lucas Rath** *lucas.rath@bmw.de*
*Institute for Mechanism Theory, Machine Dynamics and Robotics, RWTH Aachen University*
*BMW AG*

**Alexander von Rohr** *vonrohr@dsme.rwth-aachen.de*
*Institute for Data Science in Mechanical Engineering, RWTH Aachen University*

**Andreas Schultze** *andreas.schultze@bmw.de*
*BMW AG*

**Sebastian Trimpe** *trimpe@dsme.rwth-aachen.de*
*Institute for Data Science in Mechanical Engineering, RWTH Aachen University*

**Burkhard Corves** *corves@igmr.rwth-aachen.de*
*Institute for Mechanism Theory, Machine Dynamics and Robotics, RWTH Aachen University*

**Reviewed on OpenReview:** *https://openreview.net/forum?id=2iOOvQmJBK*

## Abstract

Deciding on a model structure is a fundamental problem in machine learning. In this paper we consider the problem of building a data-based model for dynamical systems from a library of discrete components. In addition to optimizing performance, we consider crash and inequality constraints that arise from additional requirements, such as real-time capability and model complexity. We address this task of model structure selection with a focus on dynamical systems and propose to search over potential model structures efficiently using a constrained combinatorial Bayesian Optimization (BO) algorithm. We propose expressive surrogate models suited for combinatorial domains and an acquisition function that can handle inequality and crash constraints. We provide simulated benchmark problems within the domain of equation discovery of nonlinear dynamical systems. Our method outperforms the state-of-the-art in constrained combinatorial optimization of black-box functions and has a favorable computational overhead compared to other BO methods. As a real-world application example, we apply our method to optimize the configuration of an electric vehicle's digital twin while ensuring its real-time capability for the use in one of the world's largest driving simulators.

## 1 Introduction

Minimizing the discrepancy between the simulated behavior of digital twins and the measured behavior of the observed system is essential in engineering applications and is commonly known as system identification (Ljung, 1998) or model learning (Nguyen-Tuong & Peters, 2011). In general, system identification comprises two interleaved subtasks, structure identification and parameter estimation. The former aims to determine the structure of model equations, while the latter focuses on finding the model parameters (Tanevski et al., 2015). While a lot of the machine learning literature focuses on the latter task, we tackle the former.

Structure identification methods explore the space of potential models and select the best-fitting representation of the dynamical system. On the one hand, structure identification can be approached by symbolic-regression methods (Bongard & Lipson, 2007; Schmidt & Lipson, 2009; Brunton et al., 2016), which explore the space

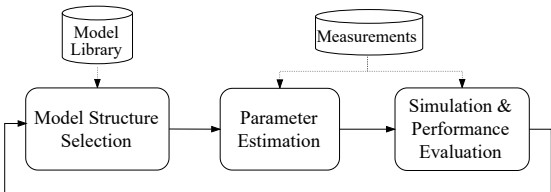

Figure 1: The process of system identification

of possible arithmetical expressions for building mathematical models and often use regularization techniques to introduce bias towards simpler models (Tanevski et al., 2020). These methods can be very general and require only the definition of the mathematical operations and functions that can compose the description of the system's behavior. On the other hand, knowledge-driven approaches (Bradley et al., 2001) require experts to encode domain-specific knowledge into model fragments, usually following known relationships, physical laws, constraints, or structural properties of the observed system. In both approaches, the idea is to break down the overall model into smaller, more manageable fragments or sub-models. This decomposition allows for a modular representation of the dynamics and enables the identification of local relationships, which is important when dealing with complex systems. In addition, both methods provide flexibility in defining the structure and form of the model fragments, which can be combined, composed, or connected in a way that reflects the relationships and interactions between the different components of the system.

In Figure 1, we depict an iterative process of system identification. In the structure identification phase, model components are selected from a model library of predefined parameterized templates that align with the system's characteristics. In the next step, regression algorithms are employed to estimate the coefficients or parameters associated with the selected templates of the model fragments. The final model is then evaluated using criteria such as goodness-of-fit, model complexity, or predictive performance. This evaluation is used as feedback to the search algorithms to select better model structures for the next iteration.

In practice, structure identification search algorithms enumerate plausible model structures and select the model templates that result in the best model evaluation (Tanevski et al., 2020). Due to the discrete nature of the problem, this task is often approached as a combinatorial optimization problem. The interplay between components disallows selecting templates independently, and an exhaustive search over all potential model structures quickly becomes infeasible due to the combinatorial explosion of the search domain. In addition, evaluating a model structure candidate involves fitting coefficients and simulating dynamical systems, which can be computationally demanding. This motivates the need for sample-efficient Bayesian optimization (BO) methods (Garnett, 2023) that are suitable for expensive-to-evaluate black-box objective functions.

Moreover, the ability to handle inequality constraints and evaluation failures during the search for the best model structure plays an important role in system identification. Inequality constraints can be used to limit the computational budget available for the model, help mitigate overfitting, and prevent models that violate physical principles. In addition, the selection of some structures might lead to invalid composition or numerically unstable simulations (Chakrabarty et al., 2021). Such failures or 'crashes' prohibit the assessment of the model performance (Bachoc et al., 2020).

We address the task of model structure selection as a combinatorial BO problem and propose a method specifically designed for models that have additional requirements that can be translated into inequality and crash constraints in the optimization. Our method efficiently searches over potential model structures and is scalable to many design parameters. The proposed method employs expressive surrogate models suited for combinatorial spaces and implements an acquisition function that handles inequality and crash constraints. While our contribution focuses on model discovery for dynamical systems, the proposed method is general and can be used for a wide variety of machine learning tasks.

We empirically evaluate the method on symbolic benchmark problems for equation discovery of nonlinear dynamical systems (Brunton et al., 2016; Mangan et al., 2017). We extend these benchmarks to include inequality constraints in the form of Lasso regularization to combat overfitting, and consider crash constraints

tackling numerical instabilities and failures. As a real-world application example, we further optimize a knowledge-driven formulation of a multibody driving simulation model.

**Our contributions** are as follows:

- We propose a general formulation for model structure selection as part of a constrained combinatorial optimization problem that considers inequality and crash constraints as a way to handle additional model requirements during the model search.
- We present `CBOSS`, a constrained combinatorial Bayesian optimization algorithm that can efficiently solve the model structure selection problem and is capable of solving combinatorial optimization problems with up to $10^{18}$ combinations. `CBOSS` employs an efficient-to-compute acquisition function and combines recently proposed surrogate models for discrete domains.
- We show empirical evidence that the proposed algorithm outperforms state-of-the-art methods on benchmark problems for equation discovery and further provide a real-world application example where `CBOSS` can build a digital twin for a driving simulator from a large library of modules, while ensuring its real-time capability.

This article continues as follows: in Section 2 we provide a general problem formulation for structure selection. The related work is discussed in Section 3. The proposed combinatorial BO method is described in Section 4. In Section 5, we investigate the performance of our method with various system identification problems and compare to other approaches.

The code for the optimizer and the benchmark problems are publicly available at https://github.com/lucasrm25/Model-Structure-Selection-CBOSS

## 2 Problem Statement

We address this structure identification problem as a constrained combinatorial optimization problem. We require a library of model templates $\mathcal{X}$ and a symbolic-regression or knowledge-driven procedure for composing the selected templates. The selection of model templates is to be parameterized with a vector of categorical decision variables denoted as $\boldsymbol{x} \in \mathcal{X}$, defined over the combinatorial domain $\mathcal{X} = \mathcal{X}_1 \times \mathcal{X}_2 \times \cdots \times \mathcal{X}_d$ with $d$ categorical decision variables with respective cardinalities $k_1, \ldots, k_d$. We introduce an objective function $f : \mathcal{X} \mapsto \mathbb{R}$ that maps the decision variables to a real value indicating the model performance. In practice, this function represents both parameter estimation and model evaluation depicted in Figure 1 and, given noisy time observations, measures the performance of the model structure specified by $\boldsymbol{x} \in \mathcal{X}$. Similarly, we define $M$ inequality constraints as $g_j : \mathcal{X} \to \mathbb{R}$ with $j \in \{1, \ldots, M\}$, and a binary equality constraint function as $h : \mathcal{X} \to \{0, 1\}$. The inequality constraint can be used, for example, to restrict the computational budget, the model complexity or as a regularization, while the binary equality constraint is used to indicate evaluation failures.

We formulate the structure selection problem as the search for the global optimizer $\boldsymbol{x}^*$ that fulfills:

$$
\begin{aligned}
\boldsymbol{x}^* = \arg\min_{\boldsymbol{x} \in \mathcal{X}} \quad & f(\boldsymbol{x}) \\
s.t. \quad & g_j(\boldsymbol{x}) \leq 0 \quad \forall j \in \{1, \ldots, M\} \\
& h(\boldsymbol{x}) = 1 \,.
\end{aligned} \tag{1}
$$

The functions $f$, $\boldsymbol{g}$, and $h$ are all expensive-to-evaluate black-box functions and can only be obtained simultaneously. The functions $f$ and $\boldsymbol{g}$ are noisy and can only be assessed when the experiment is successful, i.e.

$$
(y, \boldsymbol{c}, l) = \begin{cases} (f(\boldsymbol{x}) + \epsilon_y, & \boldsymbol{g}(\boldsymbol{x}) + \boldsymbol{\epsilon}_c, & 1) & \text{if } \boldsymbol{x} \text{ is evaluation success} \\ (\varnothing, & \varnothing & 0) & \text{if } \boldsymbol{x} \text{ is evaluation failure} \end{cases} \tag{2}
$$

where the noise $\epsilon_y \sim \mathcal{N}(0, \sigma_y^2)$ and $\boldsymbol{\epsilon}_c \sim \mathcal{N}(0, \mathrm{diag}(\boldsymbol{\sigma}_c^2))$ are i.i.d. and normally distributed. All past observations are collected in a dataset $\mathcal{D} = \{(\boldsymbol{x}^{(i)}, y^{(i)}, \boldsymbol{c}^{(i)}, l^{(i)})\}_{i \in [N]}$, where $N$ is the number of experiments.

## 3 Related Work

In this section, we review some of the existing literature of model structure selection and highlight the potential of employing combinatorial Bayesian optimization as a promising approach.

**Model Structure Selection**. Traditional methods for structure selection involve exhaustive search over the space of potential model structures, such as random search, A* search, and simulated annealing (Bertsimas & Tsitsiklis, 1993) algorithms. Others employed evolutionary algorithms (Schmidt & Lipson, 2009; Tanevski et al., 2020) or causal inference (Baumann et al., 2022). However, the main limitation of these methods is their sample efficiency, as the number of potential model structures grows exponentially with the number of discrete variables. Generally, these algorithms are not suitable for expensive-to-evaluate system identification.

A popular non-parametric class of models are kernel methods such as Gaussian processes. Here, the problem of model selection includes the choice of kernels and their composition. These problems have been tackled by greedy search (Duvenaud et al., 2013), a mixture of random walks and BO (Malkomes et al., 2016), and MCMC sampling (Gardner et al., 2017).

Mangan et al. (2017) proposed a sparse symbolic-regression structure selection approach using SINDy (Brunton et al., 2016) as a tool to down-sample parsimonious models from the combinatorially large model space.Reducing the search space allowed them to brute-force and search for the best model among the remaining ones. SINDy works by recursively performing linear regression and pruning terms with small model coefficients up to a certain threshold. Although very efficient, this method, as well as other backward-elimination algorithms, over-exploits the model space and does not guarantee to find the optimal one (Guyon & Elisseeff, 2003). In addition, SINDy cannot handle constraint problems and is not applicable to knowledge-driven problems. Another limitation is that the coefficients are pruned based on the regression results obtained by fitting a prediction model, which is not always the best strategy if the model is to be used as a simulation or auto-regressive model. BO methods are more general and allow for arbitrary cost functions, whereas SINDy decides on the structure based on the least squares cost.

**Combinatorial Bayesian Optimization**. The two main challenges in combinatorial Bayesian optimization are the development of (i) better probabilistic regression approaches that can capture the complex interaction between discrete variables and (ii) acquisition function optimizers that efficiently search the combinatorial space using the surrogates. While most methods have been defined over a mixed space of continuous and discrete variables, we focus here on the specific characteristics that are useful for discrete domains. SMAC (Hutter et al., 2011) leveraged tree-based surrogate models and used random walks to obtain a local optimum of acquisition function. This method can be applied to mixed input spaces but neglect high-order interaction between variables. BOCS (Baptista & Poloczek, 2018) encoded categorical variables in a combinatorial one-hot binary domain and used polynomial features within sparse Bayesian linear regression (Carvalho et al., 2010; Makalic & Schmidt, 2015) combined with Thompson sampling to express the interaction between variables and their effect on the objective function. They optimized the acquisition function using simulated annealing and semi-definite programming. Dadkhahi et al. (2022) extended BOCS and proposed a more compact but still complete and unique encoding that results in fewer monomials. Hase et al. (2018) and Häse et al. (2021) combined Bayesian neural networks and density kernel estimation as a surrogate for BO with categorical variables. Wan et al. (2021) proposed the Casmopolitan algorithm and used a modified Hamming kernel as part of Gaussian process regression, which defines the correlation between two inputs by their Hamming distance in the combinatorial graph. To avoid over-exploration due to high-dimensional combinatorial spaces they adapted the trust region algorithm from TURBO (Eriksson et al., 2019) to explore only locally near the best location found so far. Further, Oh et al. (2019) proposed COMBO, which used the discrete diffusion kernel built from the graph Cartesian product of discrete parameters and is able to model high-order interactions between variables. COMBO was improved by Deshwal et al. (2021) (HyBO) by using the closed-form of the discrete diffusion kernel proposed by Imre (2002). In order to tackle high-dimensional combinatorial problems Deshwal et al. (2023) propose an embedding of the high-dimensional discrete search space into a low-dimensional continuous one using dictionaries.

**Constrained Bayesian optimization** is an active area of research for continuous domains (Gardner et al., 2014; Eriksson & Poloczek, 2021; Ungredda & Branke, 2024; Marco et al., 2021) but rarely investigated for

discrete variables. Daulton et al. (2022) proposed Probabilistic Reparameterization (PR), the first method for constrained combinatorial BO, which reparameterized the discrete acquisition function optimization problem by introducing discrete probability distributions defined by continuous parameters. This allowed them to optimize the AF using gradient-based methods. We also note that other methods, such as Papalexopoulos et al. (2022), exist for constrained combinatorial optimization but only consider inexpensive and white-box constraint functions, which is not applicable to our problem.

Our proposed method extends to constrained problems and has a relatively small computational footprint. Most of the existing methods have a high computational demands and therefore scale poorly with the number of inputs and data points, since they rely on expensive MCMC methods to infer the posterior distributions. We focus on reducing the high computational overhead of the state-of-the-art methods by relying on closed-form models and AF, avoiding expensive approximations.

## 4 Methods

In the context of structure model selection, Bayesian optimization can be employed to find the optimal model structure that minimizes a chosen evaluation metric. Bayesian optimization is particularly useful when the objective function is a expensive-to-evaluate black-box function. Based on past observations, the BO algorithm constructs a probabilistic surrogate model of the objective and the constraints, typically Gaussian Processes ($\mathcal{GP}$s) (Rasmussen et al., 2006). The surrogate model is iteratively updated with new evaluations, allowing for the sequential exploration of the search space. The selection of the next promising evaluation point is guided by an acquisition function, based on the surrogate model's predictions and their uncertainty to balance the exploration-exploitation trade-off.

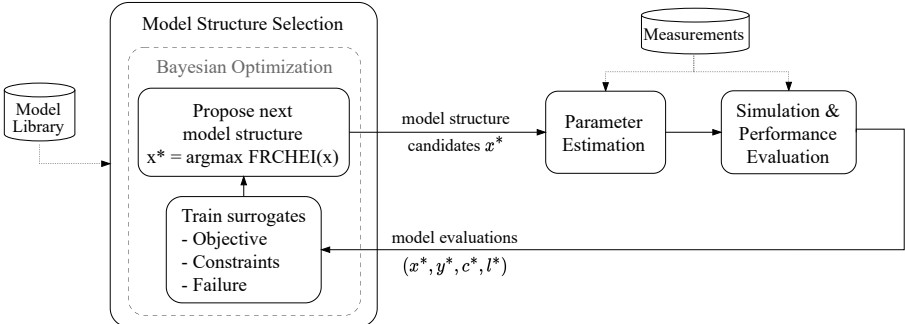

Figure 2: The proposed method for model structure selection using combinatorial Bayesian optimization. Every iteration, the model structure selection algorithm proposes a new model structure $\boldsymbol{x}$ from the model library. Using measurements, the model's parameters are estimated and the composed model performance is evaluated according to objective $y = f(\boldsymbol{x}) + \epsilon_y$, inequality constraints $\boldsymbol{c} = \boldsymbol{g}(\boldsymbol{x}) + \boldsymbol{\epsilon}_c$ and crash $l = h(\boldsymbol{x})$ function evaluations.

We call our method Constrained Combinatorial Bayesian Optimization for model Structure Selection (CBOSS). In the following subsections, we present a detailed description of this method. We define (i) probabilistic surrogate models for regression, which will approximate the black-box objective and constraint functions, (ii) a probabilistic model for classification, which will approximate the black-box binary function for failures and finally (iii) the acquisition function. An overview of the method is depicted in Figure 2 and in Algorithm 2.

### 4.1 Probabilistic Surrogate Model for Regression

We consider the setting, where objective function observations are given by $y^{(i)} = f(\boldsymbol{x}^{(i)}) + \epsilon_y^{(i)}$ and we assume the observation noise $\epsilon_y^{(i)} \sim \mathcal{N}(0, \sigma_y^2)$ to be an i.i.d. Gaussian random variable with unknown variance $\sigma_y^2$ (cf. Equation 2). After making $N$ observations, the joint distribution of observations is

$$\boldsymbol{y} \mid X, \sigma_y^2 \sim \mathcal{N}(\boldsymbol{f}(X), \ \sigma_y^2 I_N), \tag{3}$$

where $X = [\boldsymbol{x}^{(i)}, \dots, \boldsymbol{x}^{(N)}]^\top \in \mathcal{X}^N$, $\boldsymbol{f}(X) = [f(\boldsymbol{x}^{(i)}), \dots, f(\boldsymbol{x}^{(N)})]^\top \in \mathbb{R}^N$, $\boldsymbol{y} = [y^{(i)}, \dots, y^{(N)}] \in \mathbb{R}^N$ and $I_N \in \mathbb{R}^{N \times N}$ is the identity matrix. We assume the objective and constraints to be conditionally independent given $X$ and use the same setup for each $j$-th dimension of constraint function evaluations $c_j^{(i)} = g_j(\boldsymbol{x}^{(i)}) + \epsilon_{c_j}^{(i)}$. Similarly, $\boldsymbol{c}_j \mid X, \sigma_c^2 \sim \mathcal{N}(g_j(X), \, \sigma_c^2 I_N)$.

We now present a surrogate model for the expensive function $f$ as a Gaussian process prior over $f$

$$f \mid \sigma_y^2 \sim \mathcal{GP}\left(m, k\right), \tag{4}$$

where the mean function $m : \mathcal{X} \to \mathbb{R}$ and the kernel $k : \mathcal{X} \times \mathcal{X} \to \mathbb{R}$ are the crucial ingredients in $\mathcal{GP}$ regression since they define the kind of structure that will be captured by the regression model. These components are to be defined in the following subsections.

The posterior predictive distribution of a function value $f(X')$ at test points $X'$ is given analytically (Rasmussen et al., 2006)

$$\boldsymbol{f}(X') \mid X', \boldsymbol{y}, X, \sigma_y^2 \sim \mathcal{N}(\bar{\boldsymbol{\mu}}_f, \bar{\Sigma}_f) \tag{5}$$

$$\text{with} \quad \begin{aligned} \bar{\boldsymbol{\mu}}_f &= \boldsymbol{m}(X') + K(X', X)(K(X, X) + I_N \sigma_y^2)^{-1}(\boldsymbol{y} - \boldsymbol{m}(X)) \\ \bar{\Sigma}_f &= K(X', X') - K(X', X)(K(X, X) + I_N \sigma_y^2)^{-1} K(X, X') \end{aligned},$$

where $\boldsymbol{m}(X)_i = m(X_i)$ and $K(X, X')_{i,j} = k(X_i, X'_j)$.

One of the limitations of Gaussian process regression is the light tails of the predictive distribution, which are not robust to outliers (Duvenaud, 2014) and might not capture well enough the discrepancy of the surrogate model (Baptista & Poloczek, 2018). To allow for better robustness and characterization of the uncertainty in the prediction while still allowing for a closed-form posterior, we provide a formulation of surrogate regressors with hierarchical priors. A hierarchical Bayesian model is a model in which the prior distribution of some of the model parameters depends on other parameters, which are also assigned a prior. We place the following hierarchical $\mathcal{GP}$ prior over $f$

$$f \mid \sigma_y^2 \sim \mathcal{GP}\left(m, \sigma_y^2 k\right) \tag{6}$$

$$\sigma_y^2 \sim \Gamma^{-1}(\nu, \sigma_m^2), \tag{7}$$

which leads to $p(f(X), \sigma_y^2)$ being an inverse-gamma-normal distribution. To ease the derivations and the hyperparameter tuning, we use a more intuitive inverse-gamma parameterization, whose shape is $a = \nu/2$ and scale $b = \sigma_m^2 \nu/2$ (Taboga, 2017). This parameterization directly reflects the mean and the variance of this distribution since $\mathbb{E}[1/\sigma_y^2] = 1/\sigma_m^2$ and $\mathrm{Var}[1/\sigma_y^2] = 2/(\nu \sigma_m^4)$. Therefore $1/\sigma_m^2$ can be interpreted as the best guess of the regression precision, while $\nu$ expresses the degree of confidence about the precision. We follow Chen et al. (2023) and place hyperprior distributions over $\nu \sim \Gamma(\zeta_\nu, \xi_\nu)$ and $\sigma_m^2 \sim \Gamma(\zeta_{\sigma_m^2}, \xi_{\sigma_m^2})$.

The predictive prior distribution at $X$ can be calculated analytically by

$$\boldsymbol{f}(X) \mid X \sim \int p(\boldsymbol{f}(X) \mid \sigma_y^2, X) \, p(\sigma_y^2) \, d\sigma_y^2 = T(\nu, \boldsymbol{m}(X), \sigma_m^2 K(X, X)), \tag{8}$$

where $T(\nu, \boldsymbol{\mu}, \Sigma)$ is the multivariate t-distribution, with degrees of freedom $\nu$, mean $\boldsymbol{\mu}$ and scale matrix $\Sigma$.

Furthermore, the predictive posterior distribution at test points $X'$ is also given in closed-form by

$$\boldsymbol{f}(X') \mid X', \boldsymbol{y}, X \sim \int p(\boldsymbol{f}(X') \mid X', \boldsymbol{y}, X, \sigma_y^2) \, p(\sigma_y^2 \mid \boldsymbol{y}, X) \, d\sigma_y^2 = T(\bar{\nu}_f, \bar{\boldsymbol{\mu}}_f, \bar{\Sigma}_f) \tag{9}$$

$$\begin{aligned} \text{with} \quad \bar{\nu}_f &= \nu + N \\ \bar{\boldsymbol{\mu}}_f &= \boldsymbol{m}(X') + K(X', X')\left(K(X, X) + I\right)^{-1}(\boldsymbol{y} - \boldsymbol{m}(X)) \\ \bar{\Sigma}_f &= \bar{\sigma}_m^2 \left( K(X', X') - K(X', X)\left(K(X, X) + I\right)^{-1} K(X, X') \right) \\ \bar{\sigma}_m^2 &= \left( \nu \, \sigma_m^2 + (\boldsymbol{y} - \boldsymbol{m}(X))^\top \left(K(X, X) + I\right)^{-1}(\boldsymbol{y} - \boldsymbol{m}(X)) \right) / (\nu + N), \end{aligned}$$

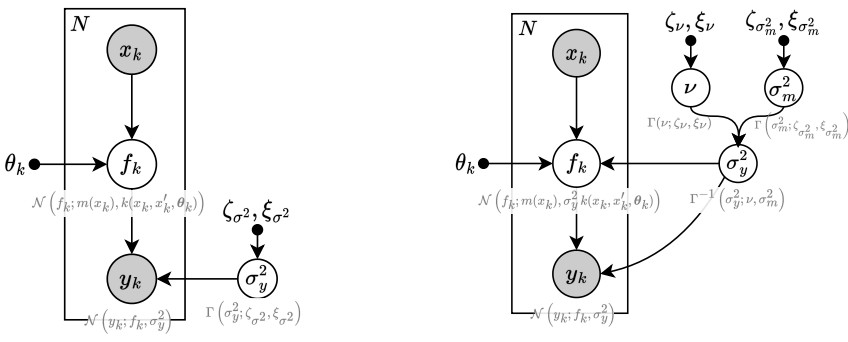

(a) Gaussian Process Regression      (b) T-Process Regression

Figure 3: Graph model of two probabilistic regression models. The array $\boldsymbol{\theta}_k$ refers to the kernel function $k$ hyperparameters.

which defines the so-called t-Process ($\mathcal{TP}$) regression framework (Shah et al., 2013; Tracey & Wolpert, 2018). Interestingly, one can show that as $N$ goes to infinity, the $\mathcal{TP}$ prior and posterior tends to the $\mathcal{GP}$ distributions. Moreover, the equations above show that the parameters $\nu, \sigma_m^2$ defined in the inverse-Gamma distribution of the prediction uncertainty $\sigma_y^2$ directly impose how heavy-tailed the predictive distribution will be. The probabilistic graph models for both $\mathcal{GP}$ and $\mathcal{TP}$ are provided in Figure 3.

### 4.1.1 Kernel Design

Kernels play a crucial role in $\mathcal{GP}$ and $\mathcal{TP}$ regression because they define the assumptions and properties of the underlying probabilistic model. The kernel defines correlations between candidates and is required for accurate and meaningful predictions. Notably, common kernels found in the literature for continuous spaces rely on a natural order of candidates, which is not given in categorical spaces. To address this limitation, many kernels specialized for categorical spaces have been recently proposed in the literature (Imre, 2002; Wan et al., 2021; Deshwal et al., 2021; Oh et al., 2019; Baptista & Poloczek, 2018; Dadkhahi et al., 2022).

In particular, the automatic relevance determination (ARD) discrete diffusion kernel became recently popular and is the discrete analog to the diffusion kernel over continuous spaces, aka the radial basis function (RBF) (Imre, 2002; Deshwal et al., 2021). This kernel defines diffusion over the entire discrete space represented by a combinatorial graph, where two nodes are connected if two configurations differ in exactly one variable. It is given by

$$k_{\text{diff}}(\boldsymbol{x}, \boldsymbol{x}') = \prod_{i=1}^{d} \left( \frac{1 - e^{-k_i \beta_i}}{1 + (k_i - 1)e^{-k_i \beta_i}} \right)^{\delta(\boldsymbol{x}_i, \boldsymbol{x}'_i)}, \tag{10}$$

where $k_i$ is the cardinality of the $i$-th categorical input variable, $\delta(\boldsymbol{x}_i, \boldsymbol{x}_j)$ is equal zero if $\boldsymbol{x}_i$ is equal $\boldsymbol{x}_j$ and equal one otherwise, and $\beta_i$ are hyperparameters that control the importance of the $i$-th discrete variable. Intuitively, this kernel is well suited for globally capturing interactions between configurations. However, one of the main limitations is that it captures similarity based on exact matches and does not consider the degree of dissimilarity between categories, which may lead to a loss of information when trying to model relationships or interactions between categorical variables.

Alternatively, Baptista & Poloczek (2018) proposed a polynomial kernel that was used in a Bayesian linear regression scheme. They observed that a universal surrogate model for a binary discrete input domain $\bar{\mathcal{X}} = \{0, 1\}^{\bar{d}}$ is given by $f(\boldsymbol{x}) = \sum_{\mathcal{S} \in 2^{\bar{x}}} \boldsymbol{\alpha}_{\mathcal{S}} \prod_{i \in \mathcal{S}} \bar{\boldsymbol{x}}_i$, where $2^{\bar{\mathcal{X}}}$ is the power set of the domain, $\boldsymbol{\alpha}_{\mathcal{S}}$ is the coefficient vector, and $\bar{\boldsymbol{x}} \in \bar{\mathcal{X}}$ is the one-hot encoding of the categorical input $\boldsymbol{x}$. In fact, this model describes all the possible configurations and becomes quickly impractical due to the exponential number of monomials.

The idea presented is to truncate monomials up to a certain degree

$$f(\boldsymbol{x}) = \boldsymbol{\alpha}_0 + \sum_j \boldsymbol{\alpha}_j \bar{\boldsymbol{x}}_{\boldsymbol{j}} + \sum_{i,j>i} \boldsymbol{\alpha}_{ij} \bar{\boldsymbol{x}}_{\boldsymbol{i}} \bar{\boldsymbol{x}}_{\boldsymbol{j}} + \sum_{i,j>i,k>j} \boldsymbol{\alpha}_{ijk} \bar{\boldsymbol{x}}_{\boldsymbol{i}} \bar{\boldsymbol{x}}_{\boldsymbol{j}} \bar{\boldsymbol{x}}_{\boldsymbol{k}} + \dots, \tag{11}$$

which can be naturally modeled as a kernel

$$k_{poly}(\boldsymbol{x}, \boldsymbol{x}') = \sigma_\alpha^2 \left( 1 + \sum_i^{\bar{d}} \bar{\boldsymbol{x}}_{\boldsymbol{i}} \, \bar{\boldsymbol{x}}_{\boldsymbol{i}}' + \sum_{i,j>i}^{\bar{d}} \bar{\boldsymbol{x}}_{\boldsymbol{i}} \bar{\boldsymbol{x}}_{\boldsymbol{j}} \, \bar{\boldsymbol{x}}_{\boldsymbol{i}}' \bar{\boldsymbol{x}}_{\boldsymbol{j}}' + \sum_{i,j>i,k>j}^{\bar{d}} \bar{\boldsymbol{x}}_{\boldsymbol{i}} \bar{\boldsymbol{x}}_{\boldsymbol{j}} \bar{\boldsymbol{x}}_{\boldsymbol{k}} \, \bar{\boldsymbol{x}}_{\boldsymbol{i}}' \bar{\boldsymbol{x}}_{\boldsymbol{j}}' \bar{\boldsymbol{x}}_{\boldsymbol{k}}' + \cdots \right), \tag{12}$$

where $\sigma_\alpha^2$ is the hyperparameter related to the prior variance of the coefficients. Notably, this kernel is more complex than the discrete diffusion kernel, especially when considering higher polynomial degrees, since it can capture higher-order interactions between categories. However, as the degree increases, the number of parameters becomes prohibitively high. By design, it also suffers from the combinatorial explosion of the discrete domain. Note also that this kernel distinguishes not only the degree of dissimilarity between different categories but the interaction between them.

In this paper, we propose to combine the strength of multiple kernels (Duvenaud, 2014) in the following way

$$k_{\text{polydiff}}(\boldsymbol{x}, \boldsymbol{x}') = \lambda \left( k_{\text{poly}}(\boldsymbol{x}, \boldsymbol{x}') \cdot k_{\text{diff}}(\boldsymbol{x}, \boldsymbol{x}') \right) + (1 - \lambda) \left( k_{\text{poly}}(\boldsymbol{x}, \boldsymbol{x}') + k_{\text{diff}}(\boldsymbol{x}, \boldsymbol{x}') \right) \tag{13}$$

where $\lambda \sim \text{Beta}(\alpha_\lambda, \beta_\lambda) \in [0, 1]$ is a hyper-parameter that controls whether the kernels should be added or multiplied together. The idea is to combine the global approximation properties of the discrete diffusion kernel with the high-order interaction and non-stationary properties of polynomial kernels. In the appendix Section A.4, we provide an ablation study of different kernels applied to our benchmark problems.

### 4.1.2 Hyperprior-parameter estimation

To optimize the hyperparameters, we resort to empirical Bayes, as suggested by Chen et al. (2023), to avoid expensive MCMC samplings. This is possible because both $\mathcal{GP}$ and $\mathcal{TP}$ regression methods have tractable marginal likelihoods. We achieve this by maximizing the marginal a-posteriori (MMAP) for the prior hyperparameters

$$\boldsymbol{\theta}^* = \arg\max_{\boldsymbol{\theta}} \; p(\boldsymbol{y} \,|\, X, \boldsymbol{\theta}) \prod_i p(\boldsymbol{\theta}_i), \tag{14}$$

where the marginal likelihood $p(\boldsymbol{y}|X, \boldsymbol{\theta}) = T(\nu, \boldsymbol{m}(X), \sigma_m^2 K(X, X) + \sigma_m^2 I_N)$ and the hyperprior distribution $p(\boldsymbol{\theta}_i)$ is assumed conditionally independent and is specifically designed for each hyperparameter $\boldsymbol{\theta}_i$. Hyperparameters without defined hyperpriors are treated as non-informative and are set to the uninformative prior $p(\boldsymbol{\theta}_i) \propto 1$. Note that we train all model hyperparameters in the model, which for the example of the $\mathcal{TP}$ model with $k_{\text{polydiff}}$ kernel consists of $\boldsymbol{\theta} = \{\nu, \sigma_m^2, \sigma_\alpha, \boldsymbol{\beta}, \lambda\}$.

### 4.2 Probabilistic Surrogate Model for Classification

When searching for possible structures of dynamical systems, it is to be expected that simulating the dynamical system might fail due to numerical instabilities. Unfortunately, it is usually not possible to delimit the regions that lead to instabilities a-priori. We model the set of feasible candidates as black-box functions and learn the failure regions from data using a probabilistic $\mathcal{GP}$ classifier. In this way, the Bayesian optimization algorithm can learn to avoid these regions while searching for the optimum. We use respectively the following prior and likelihood functions

$$h \sim \mathcal{GP}(m(\boldsymbol{x}), k(\boldsymbol{x}, \boldsymbol{x}')) \tag{15}$$

$$\boldsymbol{l} \,|\, \boldsymbol{h}(X) \sim \prod_i \text{Bernoulli}(\boldsymbol{l}_i; \; S(h(X_i))), \tag{16}$$

where $S$ is the sigmoid function, and $\boldsymbol{l}$ are previous failure observations.

We approximate the intractable Bernoulli posterior distribution $p(\boldsymbol{h}(X^*) \mid X^*, \boldsymbol{l}, X)$ with Laplacian approximation, which is simple to implement and can approximate the posterior distribution in closed-form. We use the same kernel as in Equation 13. We implement the alternating optimization procedure from Rasmussen et al. (2006) for finding the posterior mode of the Laplacian approximation (Newton-Raphson combined with Strong-Wolfe linear search) and the model hyperparameters (L-BFGS).

### 4.3 Acquisition Function

The goal of the acquisition function (AF) is to guide the search for the optimal solution in an optimization problem addressing the tension between exploitation and exploration. For the constrained problem, the AF needs to balance the potential utility of a candidate and the probability of feasibility and success. The next candidate model $\boldsymbol{x}^{(t)}$ to be evaluated at iteration $t$ is the one that maximizes the acquisition function $\alpha$ for the current model:

$$\boldsymbol{x}^{(t)} = \arg\max_{\boldsymbol{x}' \in \mathcal{X}} \ \alpha(\boldsymbol{x}') \ . \tag{17}$$

The AF needs to be optimized in every BO iteration. Generally, this optimization problem is the main computational bottleneck in BO which is why we look for an acquisition function that is cheap to evaluate and given in closed-form. In this work, we combine recent ideas in the field of Bayesian optimization and propose a new acquisition function called Failure-Robust Constrained Hierarchical Expected Improvement (`FRCHEI`), defined as follows

$$\alpha(\boldsymbol{x}') = \ \text{FRCHEI}(\boldsymbol{x}') = \ P_{\text{succ}}(\boldsymbol{x}')^{\beta_{succ} \, n/N} \cdot P_{\text{feas}}(\boldsymbol{x}')^{\beta_{feas} \, n/N} \cdot \text{HEI}(\boldsymbol{x}') \ , \tag{18}$$

where

$$P_{\text{succ}}(\boldsymbol{x}') = p(h(\boldsymbol{x}') = 1 \mid \boldsymbol{x}', \boldsymbol{l}, X) \tag{19}$$

$$P_{\text{feas}}(\boldsymbol{x}') = \prod_j^m \ p(g_j(\boldsymbol{x}') \leq 0 \mid \boldsymbol{x}', \boldsymbol{c}_j, X) \tag{20}$$

$$\text{HEI}(\boldsymbol{x}') = \mathbb{E}_{f(\boldsymbol{x}') \sim p(f(\boldsymbol{x}')|\boldsymbol{x}', \boldsymbol{y}, X)} \left[ \max\{0, y^+ - f(\boldsymbol{x}')\} \right] \ . \tag{21}$$

The idea of this AF is twofold. First, we consider the hierarchical expected improvement (`HEI`) (similar to Chen et al. (2023)). The `HEI` uses the objective function probabilistic surrogate $p(f(\boldsymbol{x}')|\boldsymbol{x}', \boldsymbol{y}, X)$ to quantify the potential improvement in the objective value over the current best feasible solution found so far $y^+ \in \boldsymbol{y}$. Samples with lower predicted mean values and higher uncertainties are more likely to be explored as they offer the potential for a better solution. `HEI` modifies the traditional `EI` by replacing the $\mathcal{GP}$ surrogate by $\mathcal{TP}$ regression, which improves the robustness against outliers while preserving the closed-form solution of the AF, which is given by

$$\text{HEI}(\boldsymbol{x}') = \bar{\sigma}_f \Big[ \underbrace{\tau^+ \Phi_T \left( \tau^+; \bar{\nu}_f, 0, 1 \right)}_{\text{exploitation}} + \underbrace{\frac{\bar{\nu}_f + (\tau^+)^2}{\bar{\nu}_f - 1} T(\tau^+; \bar{\nu}_f, 0, 1)}_{\text{exploration}} \Big] \ , \tag{22}$$

where $\tau^+ = (y^+ - \bar{\mu}_f)/\bar{\sigma}_f$, and $\phi_T$ is the CDF and $T$ is the PDF of the t-distribution, respectively. The parameters $\bar{\nu}_f, \bar{\mu}_f, \bar{\sigma}_f$ are the posterior degrees of freedom, mean and standard deviation from the posterior t-distribution at $\boldsymbol{x}'$. The derivation of Equation 22 is given in the appendix Section A.1.

Second, we avoid unfeasible regions by multiplying `HEI` with the probability of feasibility $P_{\text{feas}}$, similar to Gardner et al. (2014) and avoid instability regions by multiplying the `HEI` with the probability of success $P_{\text{succ}}$, as proposed in Chakrabarty et al. (2021). Following Hvarfner et al. (2022), we scale these probabilities with $\beta_{feas} \, n/N$ and $\beta_{succ} \, n/N$, where $n$ is the current iteration number, $N$ is the maximum number of iterations and $\beta_{feas}, \beta_{succ}$ are hyperparameters. The idea is to relax the constraints at the beginning of the optimization when we do not have much knowledge about these black-box functions. As the optimization progresses, we increasingly trust these surrogate models. Note that even though unfeasible models are never

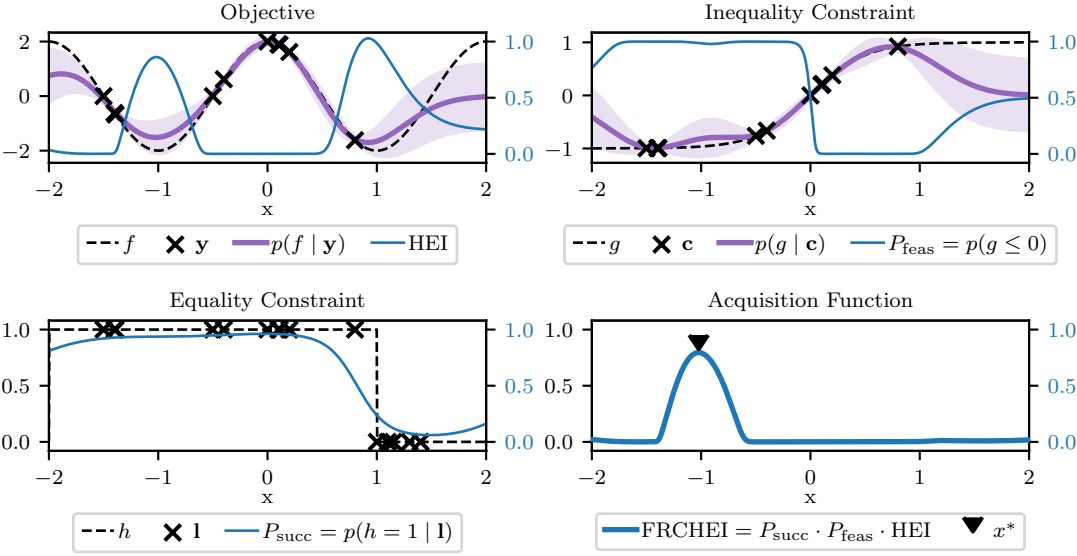

Figure 4: Illustration of the `FRCHEI` acquisition function for a one-dimensional continuous space example. The true unknown functions are depicted as dashed lines, the observations as crosses and the surrogate predictive posteriors mean and confidence region as purple lines and areas, respectively. Note that no observations could be assessed for the objective and constraint functions in failure regions $\{x : h(x) = 0\}$. The next candidate point $x^*$ is obtained by searching for the maximum of the `FRCHEI` function. Clearly, the surrogate for the constraint functions $g$ and $h$ push down regions of the expected improvement `HEI` which are likely to be unfeasible or fail. Without these two surrogates, the next candidate would be sampled on an unfeasible and failure-prone region, where the `HEI` is maximized. Note that the search space for the experiments we consider is discrete and high-dimensional, which can not be well displayed. For illustration purposes, we employed the same method but replaced the combinatorial kernels with a squared exponential.

considered the best experiment, they are still useful to improve the predictive posterior distributions (Gardner et al., 2014). An illustration of this acquisition function for a one-dimensional continuous problem example can be seen in Figure 4.

Note that as $\bar{\nu}_f \to \infty$ in Equation 22, `HEI` converges to `EI`

$$\text{EI}(\boldsymbol{x}') = \bar{\sigma}_f \left[ \tau^+ \Phi_{\mathcal{N}} \left( \tau^+; \bar{\nu}_f, 0, 1 \right) + \mathcal{N}(\tau^+; \bar{\nu}_f, 0, 1) \right] , \tag{23}$$

which is expected since t-distributions converge to Gaussian distributions as the degrees of freedom parameter $\nu \to \infty$. When using `EI` instead of `HEI` in Equation 18 we call the acquisition function `FRCEI`.

### 4.3.1 Acquisition Function Optimization

Since the input space is fully discrete, optimization of the AF can not be done directly using gradient-based methods. Instead, we use simulated annealing (Bertsimas & Tsitsiklis, 1993), which is a standard and performant method for unconstrained optimization over discrete spaces. We use the same simulated annealing approach presented in Dadkhahi et al. (2022), since it is simple, computationally efficient, and can optimize over categorical variables. For completeness, we present the algorithm again in Algorithm 1.

### 4.4 Algorithm

We present our final algorithm for black-box constrained combinatorial optimization in Algorithm 2. The algorithm requires an initial dataset $\mathcal{D}^{(1)} = \{(\boldsymbol{x}^{(i)}, y^{(i)}, \boldsymbol{c}^{(i)}, l^{(i)})\}_{i=1}^{N}$, which can be obtained by evaluating the model structure at N randomly sampled initial points $x^{(i)}$. It has been shown in recent studies (Wan et al., 2021; Deshwal et al., 2021; Müller et al., 2021; Daulton et al., 2022) that local exploitation is beneficial,

especially for high-dimensional discrete spaces. When maximizing the AF, in line 8, we make use of this idea and set the initial starting point for the `SA` optimization method as the best feasible sample evaluated so far. In addition we rerun the AF optimization twice to increase the chance of finding better candidates.

---

**Algorithm 1** Simulated annealing for categorical variables

---

1: **function** SA(objective function $f$, categorical domain $\mathcal{X}$, starting point $\boldsymbol{x}^{(0)}$, annealing scheduler $s(t)$)
2:     **for** $t = 1$ to $N$ **do**
3:         $i \sim \texttt{unif(d)}$
4:         $\boldsymbol{x}^{(t)} \leftarrow \boldsymbol{x}^{(t-1)}$
5:         $\boldsymbol{x}_i^{(t)} \sim \texttt{Softmax}\left(\{-f(\boldsymbol{x}_i = w, \boldsymbol{x}_{-i})/s(t)\}_{w \in \mathcal{X}_i}\right)$
6:     **return** $\boldsymbol{x}^{(t)}$

---

---

**Algorithm 2** Bayesian Optimization for Model Structure Selection

---

1: **function** CBOSS-FRCHEI(initial dataset $\mathcal{D}^{(1)}$)
2:     **for** $t = 1$ to $N$ **do**
3:         **Train surrogates:**
4:             $\boldsymbol{\theta}_f \leftarrow \text{MMAP}(\mathcal{D}^{(t)})$     (Equation 14)
5:             $\boldsymbol{\theta}_g \leftarrow \text{MMAP}(\mathcal{D}^{(t)})$     (Equation 14)
6:             $\boldsymbol{\theta}_l \leftarrow \text{LaplaceMode-MMAP}(\mathcal{D}^{(t)})$     (Section 4.2)
7:         **Optimize acquisition function:**
8:             $\boldsymbol{x} \leftarrow \arg\max_{\boldsymbol{x} \in \mathcal{X}} \text{FRCHEI}(\boldsymbol{x}, \boldsymbol{\theta}_f, \boldsymbol{\theta}_g, \boldsymbol{\theta}_l, \mathcal{D}^{(t)})$   (Algorithm 1 with Equation 18)
9:         $y, \boldsymbol{c}, l \leftarrow \text{EVALUATEMODELSTRUCTURE}(\boldsymbol{x})$
10:         Update data $\mathcal{D}^{(t+1)} = \mathcal{D}^{(t)} \cup \{(\boldsymbol{x}, y, \boldsymbol{c}, l)\}$
11:     **return** lowest feasible evaluation $\boldsymbol{x}^* = \arg\min_x \{y : (\boldsymbol{x}, y, \boldsymbol{c}, l) \in \mathcal{D}, \boldsymbol{c}_i(\boldsymbol{x}) \leq 0, l(\boldsymbol{x}) = 1\}$
12:
13: **function** EVALUATEMODELSTRUCTURE($\boldsymbol{x}$)
14:     Estimate model parameters, simulate and evaluate performance $(f, \boldsymbol{g}, h)$.
15:     **if** Failure **then**
16:         **return** $\{\varnothing, \varnothing, l = 0\}$
17:     **else**
18:         **return** $\{y, \boldsymbol{c}, l = 1\}$

---

## 5   Experiments and Results

In this section, we evaluate the empirical performance of `CBOSS` on a set of constrained equation discovery problems for nonlinear dynamical systems as well as a knowledge-driven configuration of a driving simulator. On the benchmark examples, we compare against two simple baselines, random sampling (`RS`) and simulated annealing (`SA`). We also compare against the state-of-the-art in constrained combinatorial optimization probabilistic reparameterization (`PR`) (Daulton et al., 2022). In our experiments, we find:

1. On the equation discovery problems, `CBOSS` is either competitive with or outperforms the other methods consistently. It always yields good equations independent of the random seed.
2. After 500 evaluations, `CBOSS` is approximately $10\times$ faster than `PR` in terms of wall clock time, highlighting the reduced computational overhead due to the closed-form inference and AF.
3. Both constrained BO methods `CBOSS` and `PR` perform especially well when finding feasible solutions is more difficult.
4. `CBOSS` is able to tune a complex driving simulator with approximately $10^{14}$ possible configurations towards a driver's preference with the constraint that the simulation must be real-time capable.

While other methods such as `BOCS`, `Casmopolitan`, `COMBO`, and `HyBO` would be interesting for this investigation their very high computational demands make them unsuitable for the the type and size of problems we

consider here. In addition, the implementations provided by the authors of these methods do not support constraints of any kind.

## 5.1 Equation Discovery for Nonlinear Dynamical Systems

In this section, we investigate system identification for the set of low-dimensional nonlinear dynamical systems shown in Figure 5. We use some of the benchmark problems and a similar learning setup from Brunton et al. (2016); Mangan et al. (2017). We investigate a range of dynamical systems: a simple nonlinear damped oscillator, a disease transmission model (SEIR), the chaotic Lorenz Oscillator, and the mean-field model for the cylinder wake in reduced coordinates. More details about these dynamical systems can be found in the appendix section A.3.

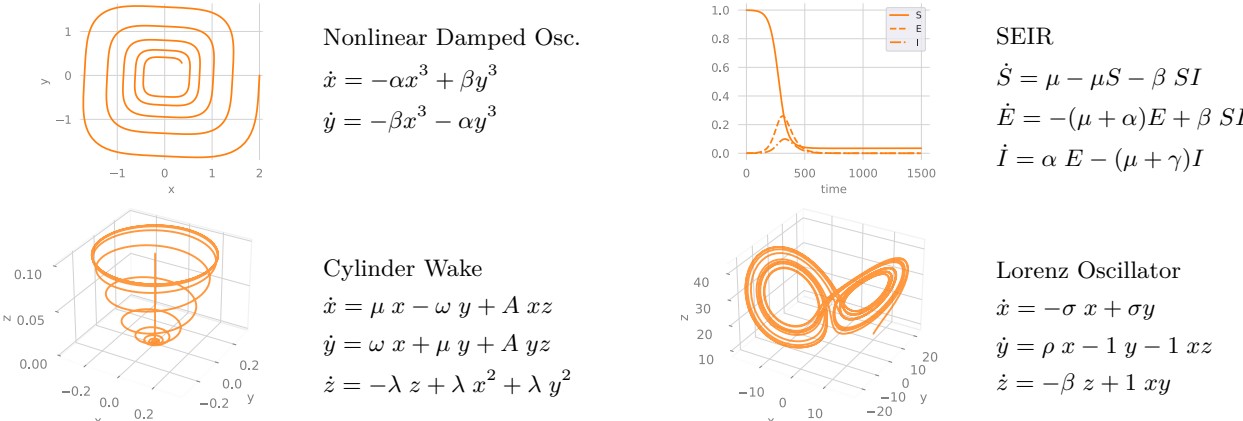

Figure 5: Equation discovery benchmark problems

Let $\boldsymbol{z}^{(t)} = \begin{bmatrix} z_1^{(t)} & \cdots & z_d^{(t)} \end{bmatrix} \in \mathbb{R}^d$ and $Z = \begin{bmatrix} \boldsymbol{z}^{(1)} & \cdots & \boldsymbol{z}^{(N)} \end{bmatrix}^\top \in \mathbb{R}^{N \times d}$ be the matrix of $N$ noisy time measurements of all $d$ states, which are available for learning the system. By differentiating the state measurements, we obtain noisy estimates of the state derivatives over time $\dot{\boldsymbol{z}}^{(t)} = \boldsymbol{f}_{\mathrm{ode}}(\boldsymbol{z}^{(t)}) + \epsilon$, with $\epsilon \sim \mathcal{N}(0, \sigma_{sn}^2)$. We seek to find the structure of the governing nonlinear differential equation $\boldsymbol{f}_{\mathrm{ode}}$, which yields the best simulation model.

A common practice in regression problems is to assume that each dimension of $\boldsymbol{f}_{\mathrm{ode}}$ can be represented by a linear combination of features $\Theta(Z) \in \mathbb{R}^{N \times n_f}$, weighted by the coefficient vector $\boldsymbol{\Xi} \in \mathbb{R}^{n_f \times d}$. These $n_f$ features compose a large library of possible nonlinear candidate functions that can represent the real system. For the benchmark problems at hand, we consider that each ODE dimension can be represented by polynomial terms up to degree $p$, which is indeed true for all the problems. In vector form equation 24 can be solved for the model coefficients $\Xi$, using a least squares estimator.

$$\underbrace{\begin{bmatrix} | & & | \\ \dot{\boldsymbol{z}}_1 & \cdots & \dot{\boldsymbol{z}}_d \\ | & & | \end{bmatrix}}_{\dot{Z}} = \underbrace{\begin{bmatrix} 1 & \boldsymbol{z}_1^{(1)} & \cdots & \boldsymbol{z}_d^{(1)} & (z_1^{(1)})^2 & \boldsymbol{z}_1^{(1)}\boldsymbol{z}_2^{(1)} & \cdots & (z_d^{(1)})^p \\ \vdots & \vdots & \ddots & \vdots & \vdots & \vdots & \ddots & \vdots \\ 1 & \boldsymbol{z}_1^{(N)} & \cdots & \boldsymbol{z}_d^{(N)} & (z_1^{(N)})^2 & \boldsymbol{z}_1^{(N)}\boldsymbol{z}_2^{(N)} & \cdots & (z_d^{(N)})^p \end{bmatrix}}_{\Theta(\boldsymbol{Z})} \cdot \underbrace{\begin{bmatrix} | & & | \\ \boldsymbol{\xi}_1 & \cdots & \boldsymbol{\xi}_d \\ | & & | \end{bmatrix}}_{\Xi} + \epsilon \quad (24)$$

Considering a large feature library enhances the flexibility and possibly the accuracy of the model up to the point that the error between the predictive and measured gradients $\dot{Z} - \Theta(Z)\Xi$ approaches zero. However, especially in the presence of noise, complex models have the potential to overfit the data leading to a deterioration of the simulation performance and even instabilities. To address this issue, we want to consider a large feature library but use structure selection to search efficiently and globally for the optimal subset of terms that construct parsimonious models, i.e., the least number of terms that significantly reduce the simulation error (Brunton et al., 2016).

Note that the number of monomials is equal to $n_t = d\, n_f = d \sum_{i=0}^{p} \binom{d+i-1}{i}$, and the total number of possible model structure combinations is given by $n_m = 2^{n_t}$, which grows exponentially with the number of monomials, which further grows factorially with the number of dimensions and the polynomial degree. An overview of the search space dimension for the benchmark problems is given in Table 1. Notably, the number of models is in the order of quintillions.

Table 1: Equation discovery benchmark problems

| Dynamical System | $d$ | $p$ | $n_t$ | $n_m$ |
|---|---|---|---|---|
| Nonlinear Damped Oscillator | 2 | 5 | 42 | 4.4e+12 |
| SEIR | 3 | 3 | 60 | 1.2e+18 |
| Cylinder Wake | 3 | 3 | 60 | 1.2e+18 |
| Lorenz Oscillator | 3 | 3 | 60 | 1.2e+18 |

We define the optimization problem in the same form as Equation 1 as follows

$$\boldsymbol{x}^* = \arg\min_{\boldsymbol{x} \in \mathcal{X}} \quad f(\boldsymbol{x}) = \log_{10} \frac{1}{N} \sum_{t}^{N} \|\boldsymbol{z}_{sim}^{(t)}(\boldsymbol{x}) - \boldsymbol{z}^{(t)}\|_1 + \lambda \log_2 \sum \boldsymbol{x} \tag{25}$$

$$s.t. \quad g(\boldsymbol{x}) = \sum_{i}^{d} \|\boldsymbol{\xi}_i(\boldsymbol{x})\|_1 - \delta \leq 0 \tag{26}$$

$$h(\boldsymbol{x}) = 1 \tag{27}$$

$$\dot{Z}_{sim} = \Theta(Z_{sim}) \cdot \Xi(\boldsymbol{x}), \tag{28}$$

where $\boldsymbol{x} \in \{0,1\}^{n_t}$ are the $n_t = d\, n_f$ binary decision variables that select which of the $n_f$ features or columns of $\Theta(Z)$ will be present in the model for each of the $d$ dimensions. $\lambda$ is a hyperparameter, $\delta$ is a constraint threshold, and $\boldsymbol{z}_{sim}(\boldsymbol{x})$ is the auto-regressive simulation trajectory of the system. The coefficients $\Xi(\boldsymbol{x}) = \begin{bmatrix} \boldsymbol{\xi}_1 & \dots & \boldsymbol{\xi}_d \end{bmatrix} \in \mathbb{R}^{n_f \times d}$ of the selected system are estimated by solving the least squares problem for each subset of $\boldsymbol{\xi}_i$, selected by $\boldsymbol{x}$. The first right-hand-side term of Equation 25 quantifies the mean absolute error between the measurements and the simulated system. The second term is a parsimony-based criterion that rewards models with fewer monomials. The inequality in Equation 26 describes the Lasso constraint and is used to restrict the model complexity. If the simulation is unstable, then $h(\boldsymbol{x}) = 0$. The final identified simulation model is given by Equation 28.

We run our method `CBOSS`, as in Algorithm 2, with batch size equal two to reduce the computation time. We investigate `CBOSS` with the two implementations of our acquisition function: `FRCHEI` and `FRCEI`. For `RS` we pick a configuration uniformly at random, and `SA` is the same as shown in Algorithm 1, with the difference that we set the objective function to infinity if the constraint is violated or the simulation is unstable. `PR` is used with the recommended settings for discrete binary optimization and batch size equal one. We provide the same randomly sampled 50 evaluations and let the optimizers run for additional 450 model evaluations. We execute each method 10 times and report mean performance and standard error.

Figure 6 shows the best feasible objective function found after a number of evaluations. The proposed methods `CBOSS-FRCHEI` and `CBOSS-FRCEI` outperform in terms of both performance and efficiency. After 500 evaluations, our methods achieve comparable or better results in a mere 3-5 hours, while `PR` takes around 30-42 hours. The `RS` and `SA` methods have a negligible computation time, and a simulation takes around 4 seconds. Note that our method runs with a batch size of two, meaning that we choose two candidates in each iteration, which halves the computational overhead. In appendix Section A.4 we show that a batch size of two is a good compromise between computation time and performance. For the SEIR and cylinder wake systems, `SA` never improves from the initial feasible solution. This is because the algorithm changes only one variable at a time, which seems to lead to either constraint violations or crashes. This highlights the need for global search methods that can escape from bad initial configurations. Both `SA` and `PR` do not perform well for the Lorenz chaotic oscillator and are stuck in local minima, where the states converge fast to a stationary point around the oscillation center (cf. Figure 7a and Figure 7b).

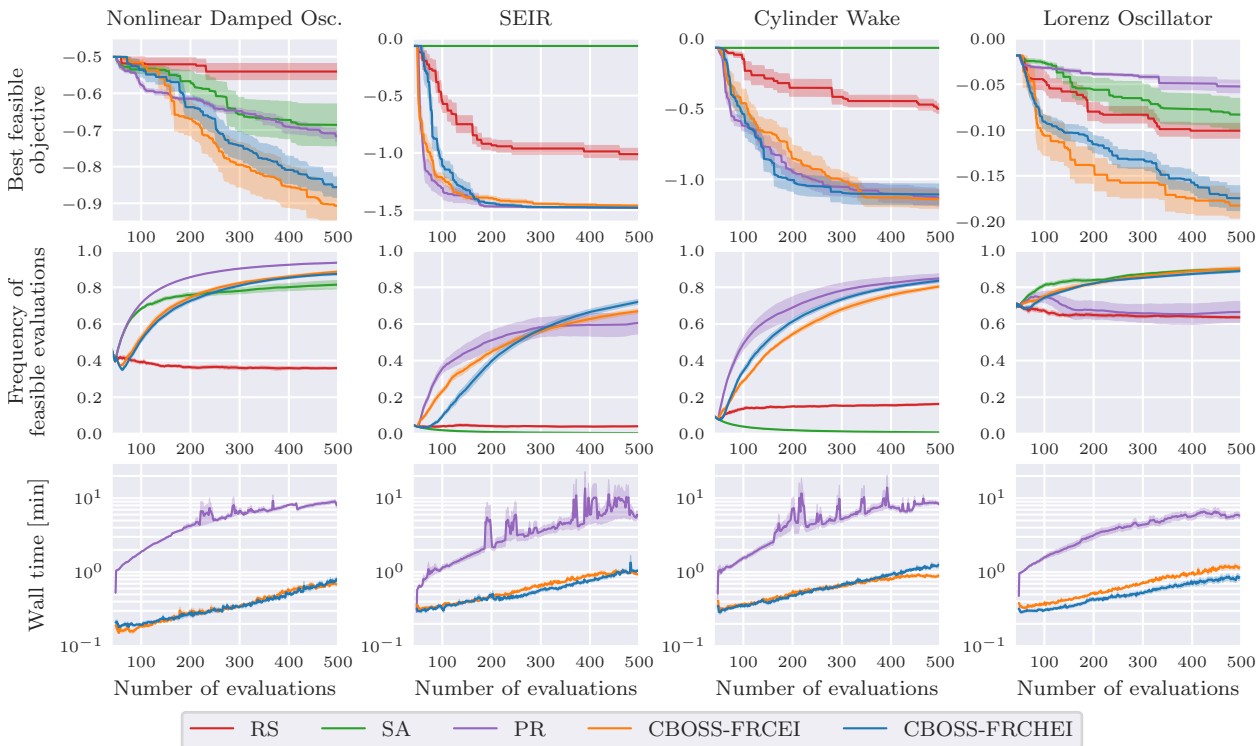

Figure 6: Best feasible objective function evaluation, wall-time per evaluation, and frequency of feasible solutions over the number of iterations for different optimization methods applied to the equation discovery problem.

In Figure 6 we show the frequency of feasible points evaluated by all methods. Our proposed AF (Equation 18) encourages early exploration and, as the classifier and constraint functions are learned, evaluates more feasible points and finds better solutions. The frequency of infeasible evaluations of the baselines further motivates the benefits of constraint BO. If the search space contains many infeasible points the surrogate model can learn these regions and concentrate its search on feasible regions. This leads to better overall solutions. `CBOSS-FRCEI` yields comparable but slightly more robust results then `CBOSS-FRCHEI`.

Figures 7a and 7b show how our method consistently outperforms others in optimizing models across multiple runs and benchmark problems, with low variance - crucial for expensive one-time optimization tasks. Even the pure exploratory policy underperforms. A balanced method like `CBOSS` strikes the right trade-off between exploration and exploitation, effectively navigating model structures and avoiding local minima.

The appendix Section A.4 provides ablation studies with further insights and analysis of the proposed method. We investigate the performance of `CBOSS` when not accounting for failures in the AF. We also consider the effect of using different kernels and show that the proposed combination of popular kernels outperforms others for these benchmark problems. Finally, we inspect the effect of varying the batch size.

## 5.2 Real-time Multibody Dynamics Model Optimization

In this experiment, we aim to optimize a digital twin configuration of an electric vehicle for a driving simulator, which is crucial for developing driving dynamics and advanced driver assistance systems. Driving simulators minimize the need for prototypes and shorten development cycles by enabling early testing of car development under various realistic conditions before road testing has even started (BMW AG, 2020). Achieving realistic driving dynamics in simulators requires hardware capable of replicating the dynamic range and real-time motion cueing, along with precise digital-twin vehicle models. Companies have invested significantly in cutting-edge simulators to meet these needs, making the design of accurate simulation models a pivotal

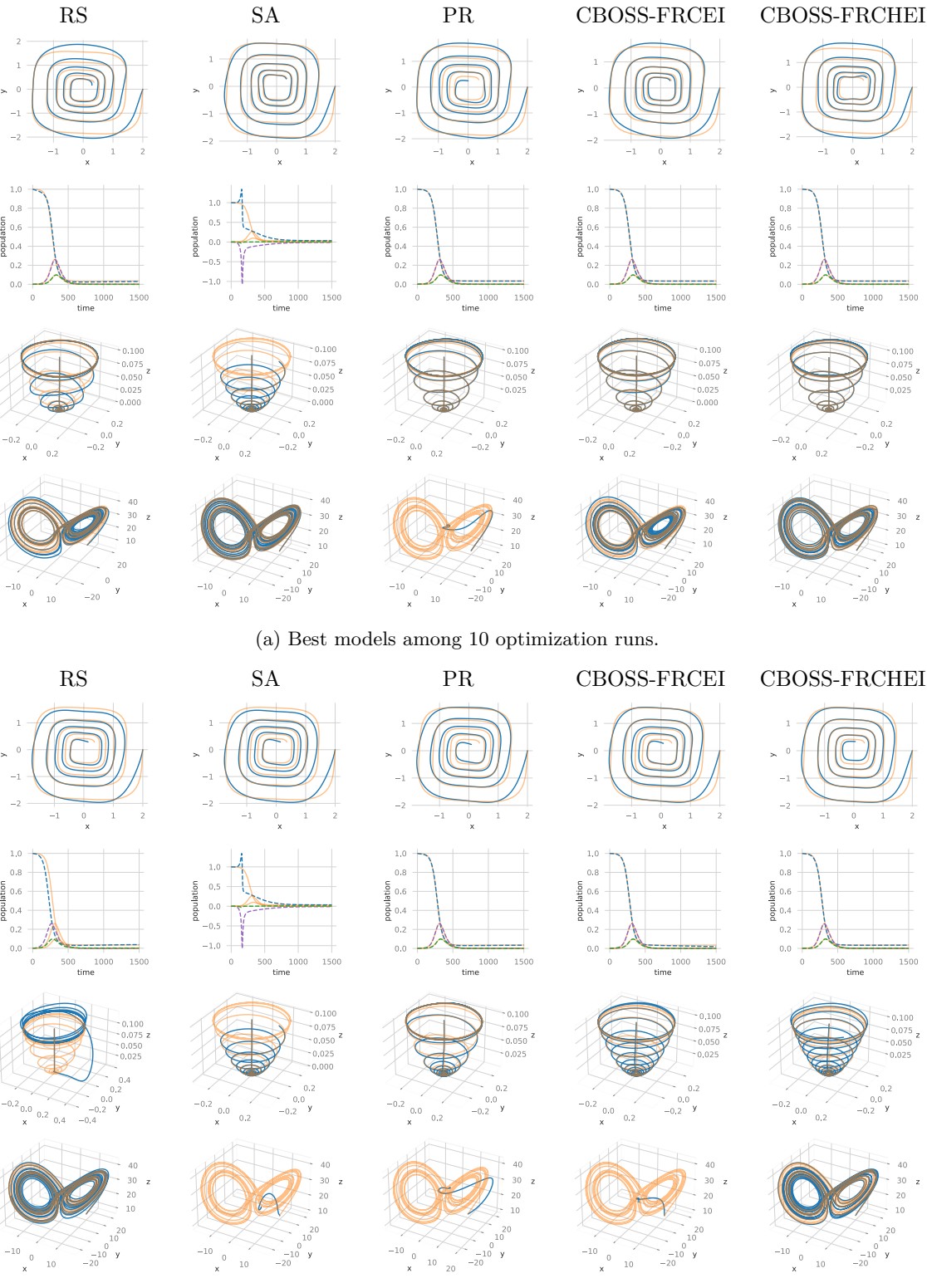

(a) Best models among 10 optimization runs.

(b) Worst models among 10 optimization runs.

Figure 7: Comparison between measurements (orange) depicted without noise and simulated trajectories of the best feasible model after optimization (blue).

element in virtual vehicle development. Figure 8 shows the high-fidelity simulator at the BMW AG driving simulation site.

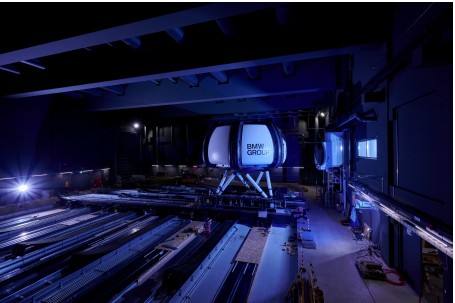 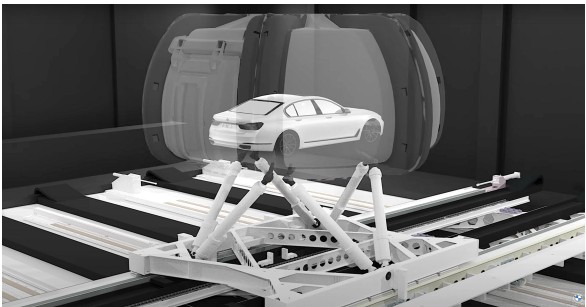

Figure 8: High-fidelity driving simulator with 9 degrees of freedom, motion area of nearly 400 square meters and a peak electrical power required up to 6.5 MW. Source BMW AG (2020).

The digital twin is a multibody dynamics model, which consist of multiple entities (bodies, joints, force and control elements, etc.) (Featherstone, 2014). Every entity has multiple possible implementations, and the goal is to choose a suitable one from a library while retaining the overall model real-time capability. For different use-cases of the driving simulator, different implementations are necessary. This requires the digital twin to be adapted on a regular basis. We aim to automate this process using our proposed BO method. For example, a damper can be implemented as a simple lookup table or as a differential equation. While a complex damper model improves simulation accuracy on a bumpy road, a simple lookup table is sufficient for longitudinal dynamics maneuvers. In summary, the major challenge in system identification for the driving simulator boils down to selecting the appropriate templates for each entity in order to optimize model simulation accuracy while simultaneously constraining the computational complexity to enable real-time simulations. However, this task is not straightforward. Model designers face an overwhelming number of potential configurations arising from the combinatorial explosion when selecting templates. Moreover, adding complexity to one component doesn't always enhance model accuracy or proportionally increase overall the computational complexity.

We investigate automatic template selection for knowledge-driven structure selection with our proposed combinatorial BO method `CBOSS`. We model the vehicle using the Simpack[1] simulation software and parameterize the model with 46 categorical variables that alter the template of different components. The vehicle for one of the configurations is depicted in Figure 9a. All the template coefficients were identified in a preliminary phase, so there is no need for a parameter estimator. The discrete design space $\mathcal{X}$ is defined as follows:

- 5 binary variables switch between rigid and flexible formulation of bodies.
- 6 categorical variables switch the number of modes used in the model-order-reduction of various flexible bodies.
- 2 binary variables switch the tire contact model approach.
- 4 binary variables switch between rigid (joint) and compliant (force element) motor mounts, suspension rods and stabilizer bushings.
- 29 binary variables switching the complexity of lookup tables (linear and nonlinear) representing the stiffness of various compliant bushing elements.

The structure identification as a constrained combinatorial optimization problem is defined as

$$\boldsymbol{x}^* = \arg\min_{\boldsymbol{x} \in \mathcal{X}} \quad f(\boldsymbol{x}) \tag{29}$$
$$s.t. \quad g(\boldsymbol{x}) = \mathrm{RTI}_{\max}(\boldsymbol{x}) - 1 \leq 0$$
$$h(\boldsymbol{x}) = 1 \,,$$

where the function $f(\boldsymbol{x}) : \mathcal{X} \to \mathbb{R}$ defines the model performance. In our experiment, the objective function reflects the preference of expert drivers for one model over another. The real-time-index (RTI), is defined as

---

[1]https://www.3ds.com/products/simulia/simpack

the ratio of the time required to advance one time step in the simulation divided by the simulation time step size. Real-time models must have an RTI below 1 in 99% after the first 2 seconds of 'warming-up'. Simulation failures arise from certain input combinations that result in numerical instabilities or in an invalid model, for instance, when it induces kinematic loops that can not be handled by an ODE solver.

We provide the evaluation of 30 initial random samples for the optimizer, with a total evaluation budget of 300 evaluations and run the optimization twice. Figure 9b shows the simulation results for the problem at hand. It can be seen that the proposed method finds good real-time capable configurations. Figure 9c depicts the `RTI` signal for the first 10 seconds of simulation for the best feasible model found during optimization among the two runs. We leave the detailed objective and subjective evaluation of the optimized simulation model for future work.

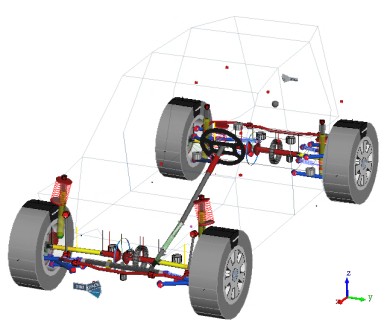
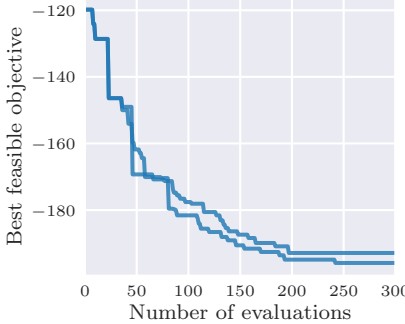
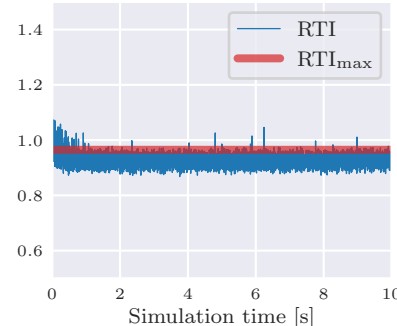

(a) 3D view of the multibody dynamical model representing the vehicle.

(b) Progress of the 2 optimization runs for the best feasible solution over number of model evaluations.

(c) RTI measurement of the final best vehicle found during optimization.

Figure 9: Optimization results for the multibody dynamics problem with the `CBOSS-FRCHEI` optimizer.

## 6 Conclusion and Outlook

In this paper, we tackle structure identification in dynamical systems under constraints and potential failures. We introduce constraints to manage computational resources and prevent overfitting. Our approach utilizes combinatorial Bayesian optimization to search for optimal model structures efficiently. We present `CBOSS`, a combinatorial Bayesian optimization algorithm that effectively manages constraints and failures with low computational overhead.

We encode the choice of the structure of the dynamical system using binary and categorical decision variables. Our method handles these discrete inputs by employing kernels specific for discrete inputs. We combine recent ideas in kernel design and show that the proposed kernel outperforms state-of-the-art kernels. Our method handles the black-box constraint and failure functions by learning them with Bayesian regression and classification methods and therefore learns to avoid these regions during search. Finally, we focus on scalability up to a large number of discrete decision variables and make design choices that favour the run time. Our surrogate models and acquisition function are evaluated in closed-form. This allows our method to optimize problems with up to $10^{18}$ possible combinations.

We provide benchmark problems in the field of symbolic-regression that provide evidence that our method outperforms other methods for system identification of a variety of nonlinear dynamical systems, such as disease models, oscillators and chaotic systems. In addition, we provide a complex real-world application example of knowledge-driven system identification where the choice of templates of a multibody dynamical system of an electric vehicle are optimized for accuracy, real-time capabilities and numerical robustness. As the proposed method is not specific to the presented applications, it is in principle applicable to other constrained combinatorial problems. Investigating its potential and performance on problems, such as network structure optimization (Du et al., 2022), wait-and-judge scenario optimization (Campi & Garatti, 2018), and neural structure optimization with pure categorical variables are interesting topics for future research.

Other promising future research directions are improved surrogate models that capture higher-order interactions of variables or leverage the structure of dynamical systems. In addition, there is often a lot of additional prior knowledge about the problem at hand that can be incorporated. For example, expert beliefs (Hvarfner et al., 2022) or qualitative knowledge about the interaction of terms which can be modeled as graphs.

## Acknowledgements

Simulations were performed with computing resources granted by RWTH Aachen University under project `rwth1409`.

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

# A Appendix

## A.1 Constrained Hierarchical Expected Improvement

In this section, we provide the remaining ingredients to derive the closed-form expression for the acquisition function used: the failure-robust constrained hierarchical expected improvement (FRCHEI) as defined in Equations 18-22.

Let the hierarchical predictive posterior for the objective and for each $i$-th constraint function at a test point $\boldsymbol{x}'$ be defined as $\tilde{f}(\boldsymbol{x}') = f(\boldsymbol{x}') \mid \boldsymbol{x}', \boldsymbol{y}, X \sim T(\bar{\nu}_f, \bar{\mu}_f, \bar{\sigma}_f^2)$ and $\tilde{g}_i(\boldsymbol{x}') = g_i(\boldsymbol{x}') \mid \boldsymbol{x}', \boldsymbol{y}, X \sim T(\bar{\nu}_{g_i}, \bar{\mu}_{g_i}, \bar{\sigma}_{c_i}^2)$ respectively. The definition of the constrained hierarchical improvement function is inspired by Gardner et al. (2014) and is defined as

$$\mathrm{CHI}(\boldsymbol{x}') = \Delta(\boldsymbol{x}') \max\{0, y^+ - \tilde{f}(\boldsymbol{x}')\}, \tag{30}$$

where $y^+$ is the lowest feasible objective function observed so far and $\Delta(\boldsymbol{x}') \sim \mathrm{Bernoulli}(\gamma(\boldsymbol{x}')) \in \{0, 1\}$ is the feasibility indicator function with parameter

$$P_{\mathrm{feas}}(\boldsymbol{x}') := \gamma(\boldsymbol{x}') \tag{31}$$
$$= \mathbb{E}\left[\Delta(\boldsymbol{x}')\right] \tag{32}$$
$$= p(\tilde{g}_1(\boldsymbol{x}') \leq 0, \ldots, \tilde{g}_m(\boldsymbol{x}') \leq 0) \tag{33}$$
$$= \prod_{i=1}^{m} p(\tilde{g}_i(\boldsymbol{x}') \leq 0) \tag{34}$$
$$= \prod_{i=1}^{m} \int_{-\infty}^{0} p(g_i(\boldsymbol{x}') \mid \boldsymbol{x}', \boldsymbol{y}, X) \, dg_i(\boldsymbol{x}') \tag{35}$$
$$= \prod_{i=1}^{m} \Phi_T(0; \bar{\nu}_{g_i}, \mu_{g_i}, \sigma_{c_i}^2). \tag{36}$$

Note that from Equation 33 to Equation 34 we assumed the simplest case, where are constraints are conditionally independent given $\boldsymbol{x}'$. In addition, Equation 35 is a simple product of univariate cumulative density of the t-distribution $\Phi_T$ evaluated at 0, which can be calculated analytically for any $\boldsymbol{x}'$.

The constrained hierarchical expected improvement is obtained by taking the expectation of Equation 30:

$$\mathrm{CHEI}(x') = \mathbb{E}\left[\mathrm{CHI}(\boldsymbol{x}')\right] \tag{37}$$
$$= \mathbb{E}\left[\Delta(\boldsymbol{x}')\right] \mathbb{E}\left[\max\{0, y^+ - \tilde{f}(\boldsymbol{x}')\}\right] \tag{38}$$
$$= P_{\mathrm{feas}}(\boldsymbol{x}') \, \mathrm{HEI}(\boldsymbol{x}'), \tag{39}$$

where $P_{\mathrm{feas}}(\boldsymbol{x}')$ is defined as in Equation 31 and HEI can be obtained with the help of the reparameterization trick (Tracey & Wolpert, 2018) with $\tau \sim T(\bar{\nu}_f, 0, 1)$ and $\tau^+ = (y^+ - \bar{\mu}_y)/\bar{\sigma}_y$:

$$\mathrm{HEI}(\boldsymbol{x}') = \mathbb{E}_{f(\boldsymbol{x}') \sim p(f(\boldsymbol{x}')|\boldsymbol{x}', \boldsymbol{y}, X)}\left[\max\{0, y^+ - f(\boldsymbol{x}')\}\right] \tag{40}$$
$$= \int_{-\infty}^{y^+} (y^+ - f(\boldsymbol{x}')) \, T(f(\boldsymbol{x}') \, ; \bar{\nu}_f, \bar{\mu}_f, \bar{\sigma}_f^2) \, df(\boldsymbol{x}') \tag{41}$$
$$= \int_{-\infty}^{\tau^+} (y^+ - \bar{\mu}_f - \bar{\sigma}_f \tau) \, T(\tau \, ; \bar{\nu}_f, \bar{\mu}_f, \bar{\sigma}_f^2) \, d\tau \tag{42}$$
$$= (y^+ - \bar{\mu}_f) \, \Phi_T\left(\tau^+; \bar{\mu}_f, 0, 1\right) - \bar{\sigma}_f \int_{-\infty}^{\tau^+} \tau \, T(\tau \, ; \bar{\nu}_f, \bar{\mu}_f, \bar{\sigma}_f^2) \, d\tau \tag{43}$$
$$= \bar{\sigma}_f \, \tau^+ \, \Phi_T\left(\tau^+; \bar{\nu}_f, 0, 1\right) + \bar{\sigma}_f \left(\frac{\bar{\nu}_f}{\bar{\nu}_f - 1}\right) \left(1 + \frac{\tau^{+2}}{\bar{\nu}_f}\right) T(\tau^+ \, ; \bar{\nu}_f, 0, 1) \tag{44}$$
$$= \bar{\sigma}_f \left[\tau^+ \, \Phi_T\left(\tau^+; \bar{\nu}_f, 0, 1\right) + \frac{\bar{\nu}_f + \tau^{+2}}{\bar{\nu}_f - 1} \, T(\tau^+ \, ; \bar{\nu}_f, 0, 1)\right]. \tag{45}$$

The other minor modifications to turn `CHEI` into `FRCHEI` are defined in Section 4.3.

## A.2 Hyperparameters

For each optimizer, we use the same hyperparameters across all experiments. These hyperparameters have been hand-tuned to reasonable values so structure selection works well for a wide variety of problems. Table 2 shows the hyperparameters used for the optimizers investigated in this work. For the `SA` algorithm, we use the same temperature scheduler as the one used for AF optimization in `CBOSS`. The `PR` method has been used with the default parameter values suggested by Daulton et al. (2022). We set the number of MC samples to 256 as it provides a good tradeoff between performance and wall time. We follow their recommendation and disable trust region over the Boolean variables.

Table 2: Optimizer hyperparameters used for all experiments. The upper-script $^{(0)}$ denotes the initial value used for hyperparameter optimization. In `CBOSS`, we use the same regression hyperparameters to learn the objective and inequality constraint functions.

| Alg. | Hyperparameter | Reference | Values |
|------|----------------|-----------|--------|
| CBOSS | $\mathcal{TP}$ regression : $\sigma_m^2$ hyperprior | Equation 7 | $\sigma_m^2 \sim \Gamma(2, 15),\ \ \sigma_m^{2\,(0)} = 0.15$ |
| CBOSS | $\mathcal{TP}$ regression : $\nu$ hyperprior | Equation 7 | $\nu \sim \Gamma(2, 0.5),\ \ \nu^{(0)} = 2.5$ |
| CBOSS | $\mathcal{GP}$ regression : $\sigma_y^2$ hyperprior | Equation 3 | $\sigma^2 \sim \Gamma(2, 15),\ \ \sigma^{2\,(0)} = 0.15$ |
| CBOSS | kernel : discrete diffusion | Equation 10 | $\beta_i^{(0)} = 1$ for $i=1\dots d$ |
| CBOSS | kernel : polynomial+diffusion | Equation 13 | $\lambda \sim \text{Beta}(1.5, 1.5),\ \ \lambda^{(0)} = 0.5$ |
| CBOSS | AF exponents | Equation 18 | $\beta_{\text{feas}} = 20,\ \beta_{\text{succ}} = 5$ |
| CBOSS | AF optimizer : SA scheduler | Algorithm 1 | $s(t) = 0.5 \cdot 0.95^t$ |
| CBOSS | AF optimizer : SA max. iterations | Algorithm 1 | $N = 100$ |
| CBOSS | AF optimizer : SA reruns | Algorithm 2 | 4 |
| SA | SA scheduler | Algorithm 1 | $s(t) = 0.5 \cdot 0.99^t$ |
| PR | MC samples | Daulton et al. (2022) | 256 |

In Figure 10 we show the hyperpriors and initial values used in `CBOSS`. For all the other hyperparameters, we use uninformative hyperpriors. Given $\mathbb{E}[1/\sigma_y^2] = 1/\sigma_m^2$ and $\text{Var}[1/\sigma_y^2] = 2/(\nu\sigma_m^4)$, we interpret $1/\sigma_m^2$ as the estimated regression precision and $\nu$ as the confidence in this precision. We thus select Gamma hyperpriors for $\sigma_m^2$ and $\nu$ with positive support. Before training, we normalize the target outputs to a mean of zero and a standard deviation of one, and set a Gamma hyperprior for $\sigma_m^2$ with a mode at 0.2, indicating 20% of the data variance, while assigning a low likelihood for $\sigma_m^2$ exceeding half the variance. We assign a distinct Gamma hyperprior to $\nu$ to reflect plausible values for normalized data. For the $\lambda$ variable in Equation 13, which balances kernel multiplication and summation, we choose a Beta distribution with a mode at 0.5 and a broad density range. These hyperpriors represent our beliefs about the latent variables' true values and guide the optimizer in learning these hyperparameters from the data, as detailed in Equation 14.

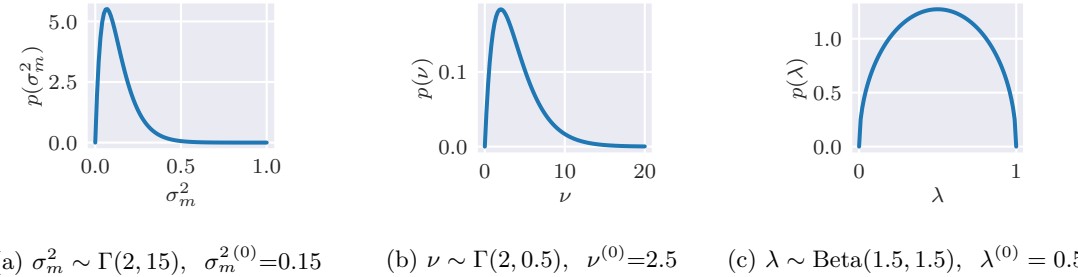

(a) $\sigma_m^2 \sim \Gamma(2, 15),\ \ \sigma_m^{2\,(0)}{=}0.15$     (b) $\nu \sim \Gamma(2, 0.5),\ \ \nu^{(0)}{=}2.5$     (c) $\lambda \sim \text{Beta}(1.5, 1.5),\ \ \lambda^{(0)} = 0.5$

Figure 10: Informative hyperpriors used in `CBOSS`.

Figure 11 shows the modified probability of feasibility and success, as seen in Equation 18. It can be seen that the exponents $\beta \, n/N$ relax the constraints at the beginning of the optimization when there is no much knowledge about the inequality and equality black-box functions. As the optimization progresses, we increasingly trust these surrogate models and increase the impact of the probability of feasibility in the AF.

Figure 12 shows the simulated annealing scheduler $s(t)$ used in Algorithm 1 as a function of the number of iterations. In this paper, we use the exponential decay scheduler $s(t) = T_0 \, d^t$, where $T_0$ is the initial temperature, $d$ is the temperature decay and $n$ is the iteration number. We calculate $d$ such that the temperature decays from $T_0 = 0.5$ to the final temperature of $T_e = 0.005$ at the final iteration.

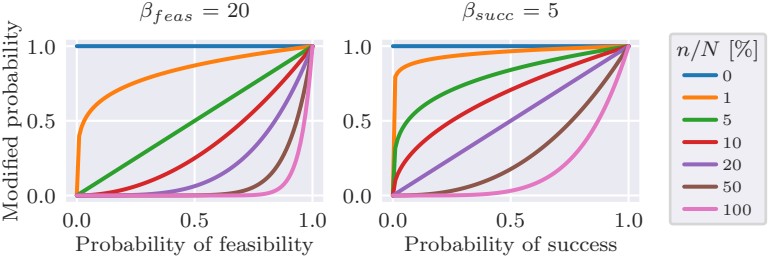
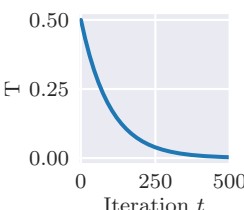

Figure 11: Modified probability of feasibility and success $P_{mod} = P^{\beta \, n/N}$ as a function of the probability of feasibility $P = P_{feas}$ or success $P = P_{succ}$, the current iteration $n$ and total number of iterations $N$.

Figure 12: SA scheduler with initial temperature $T_0 = 0.5$ and decay $d = 0.99$.

## A.3 Experiment Details

Table 3 provide the experiment details required to reproduce each benchmark problem in Section 5.1. The parameters $\boldsymbol{\theta}_{ode}$ refer to the differential equations for each benchmark problem, as depicted in Figure 5. The available noisy measurements for parameter estimation and model evaluation consists of a single simulation run, starting at an initial state $x(t_0 = 0)$, simulated with a fixed simulation time step size $\Delta_t$ up to the stop time $t_f$. The state observations are corrupted equally for each dimension with a zero-mean Gaussian noise with standard deviation $\sigma_{sn}$. All measurements have been filtered using the Total Variation Regularized Denoising technique (Chartrand, 2011; Van Breugel et al., 2022) with the same regularization parameter $T\gamma = 0.01$ and number of iterations $T_i = 10$.

Table 3: Benchmark problem parameters

| Dynamical System | $\boldsymbol{\theta}_{ode}$ | $x(t_0)$ | $\Delta_t$ | $t_f$ | $\sigma_{sn}$ |
|---|---|---|---|---|---|
| Nonlinear Damped Osc. | $\alpha=0.1, \beta=1.75, \gamma=1/2$ | $[2, 0]$ | 0.01 | 35 | 0.1 |
| SEIR | $\mu=1e{-}5, \alpha=1/5, \beta=1.75, \gamma=1/2$ | $[0.9995, 4e{-}4, 1e{-}4]$ | 0.1 | 150 | 0.01 |
| Cylinder Wake | $\omega=1, \mu=0.1, A={-}1, \lambda=1.$ | $[0.001, 0, 0.1]$ | 0.1 | 100 | 0.01 |
| Lorenz Oscillator | $\sigma=10, \rho=28, \beta=8./3$ | $[10, 10, 10]$ | 0.01 | 20 | 1 |

All the benchmark problems have been simulated with a fixed-step RK45 solver, where the 5th stage has been used to detect numerical instabilities, which stops the simulation and reports the failure to the main program. We provide a self-implementation of this solver that performs simulations in parallel and is useful to speed up computation when many simulations are executed with the same integration time steps but with different model parameters.

The benchmark experiments ran on Intel Xeon Platinum 8160 Processors "SkyLake" at 2.1 GHz, on 4 isolated physical cores. The optimization and the simulations of the multibody dynamical problem were performed on a 2x Intel Xeon Gold 6256 3.6Hz computer, running a RedHawk Linux RTOS (by Concurrent Real-Time). The first 12 cores were dedicated to the optimizer, while the remaining 12 cores of the second CPU socket were shielded and dedicated to simulations.

## A.4 Ablation Studies

In this section, we provide ablation studies that reinforce the design choices made for the proposed algorithm.

Figure 13 shows simulation trajectories for the best feasible and stable models found over the progress of the optimization for the `CBOSS-FRCHEI` method. It can be seen that the `CBOSS-FRCHEI` algorithm quickly improves the model simulation performance. The second image from left to right shows the simulated trajectory of the best model found after 163 model structure evaluations, which is already very close to the measurement trajectory. This illustrates well the efficiency of the proposed method in terms of the number of evaluations needed for system identification.

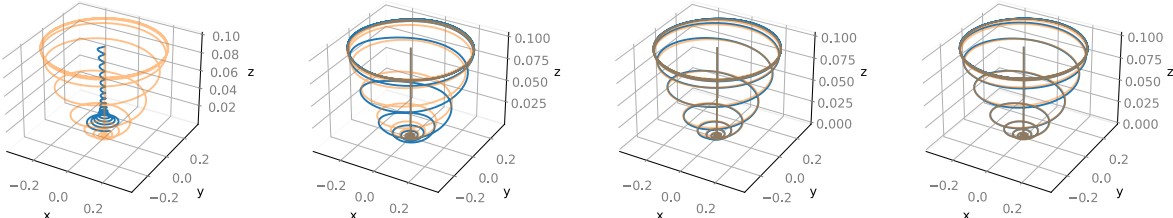

Figure 13: Comparison between measurements (orange) depicted without noise and simulated trajectories (blue) for the best feasible model found in the course of optimization for the Cylinder Wake problem and `CBOSS-FRCHEI`. From the left to the right are depicted the models at iteration 8, 163, 200 and 436, respectively.

In Figure 14, we compare `CBOSS-FRCHEI` to `CBOSS-FRCEI` and `CBOSS-CHEI`, two variants of the proposed method. The first is similar to `CBOSS-FRCHEI` but uses a $\mathcal{GP}$ instead of a hierarchical $\mathcal{TP}$ prior for the regression surrogates. The second does not account for failures in the AF. It can be seen that `FRCHEI` outperforms `CHEI` for all benchmarks, which agrees with Chakrabarty et al. (2021) and is an indication that learning failure regions improves the optimizer. As expected, `CBOSS-FRCHEI` has a higher success rate than `CBOSS-CHEI` for all benchmark problems. Evaluating failures is a waste of resources because no value for the cost function is obtained, and the candidate configuration is never considered the best experiment. In addition, this data point does not contribute to training the surrogates due to the missing target value. For this reason, the optimizer might get stuck into failure regions, where the objective function surrogate considers there is a substantial expected improvement.

It can be further seen in Figure 14 that `FRCEI` provides similar performance and sometimes slightly outperforms `FRCHEI`. The idea behind hierarchical priors is to be able to better describe the model discrepancy towards outliers while preserving the closed-form solution of the acquisition function. One side effect of this approach is the inflation of the predictive posterior uncertainty that might lead to over-exploration of the design space, which is critical for Bayesian optimization in high-dimensional discrete problems. Nevertheless, we still offer a $\mathcal{TP}$ regression formulation in our framework since it it does not increase the implementation complexity and computation time, and may work better for other problems.

Moreover, we investigate the effect of varying the batch size in Figure 15. We choose the batch size equal 2, since it provides a good trade-off between optimization performance and wall time. Throughout this paper, all the other experiments with `CBOSS` have been conducted with batch size equal 2. It is also important to note that we do not employ more involving batch evaluation strategies, such as Kriging believer strategy (Ginsbourger et al., 2010), which can potentially increase even more the performance at the cost of increasing the wall time.

Finally, in Figure 16, we compare the performance of the proposed method with different kernels. Evidently, combining the degree 2 polynomial kernel with the discrete diffusion kernel as in Equation 13 provides the best results. Note that the wall time for the polynomial kernel scales almost linearly with the number of samples but is expensive due to the number of multiplications that arise due to the large number of features resulting from the second-order polynomial combination of inputs. The discrete diffusion kernel results in a lower wall time at low number of evaluations but increases at a higher rate. This is due to the complexity of calculating pair-wise delta functions between all inputs in the dataset. In terms of computational complexity,

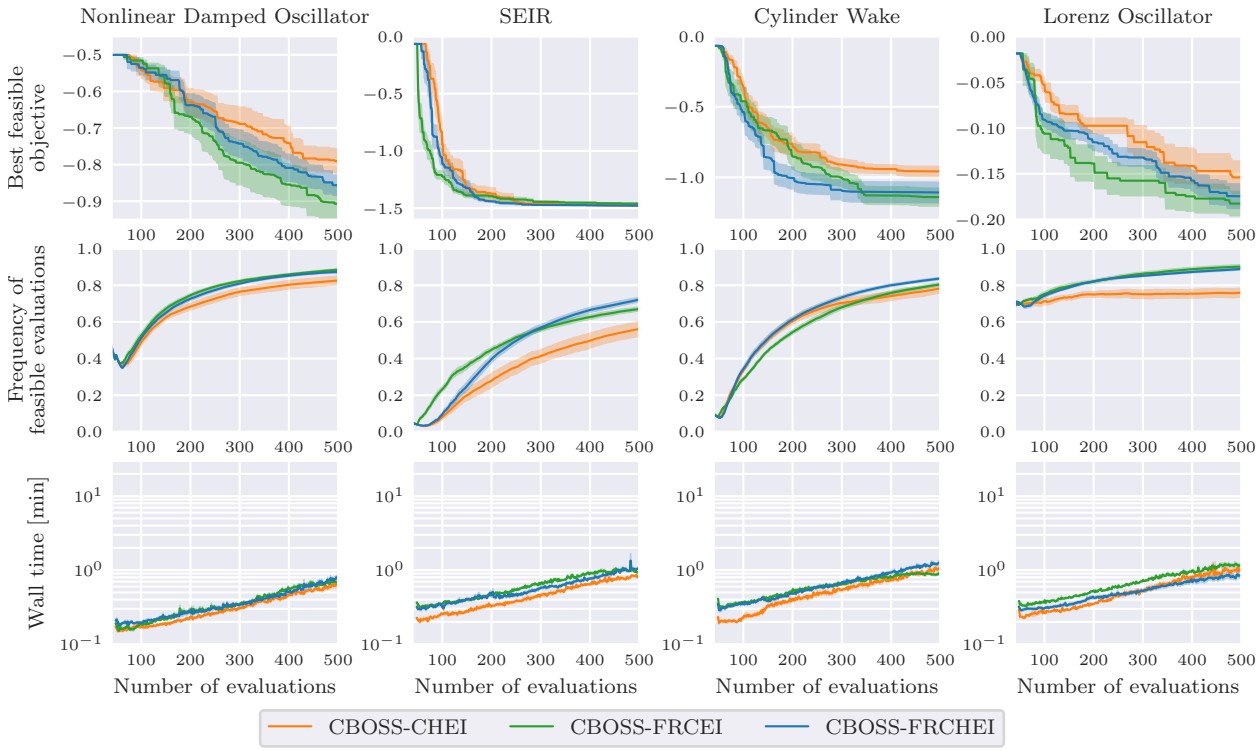

Figure 14: Optimization results for different variants of `CBOSS`. `CBOSS-CHEI` is the variant that does not handle failures, and `CBOSS-FRCEI` is the variant that implements a $\mathcal{GP}$ instead of $\mathcal{TP}$ regression.

the final performance should be evaluated individually for the problem at hand and depends on the number of samples and the number of input dimensions.

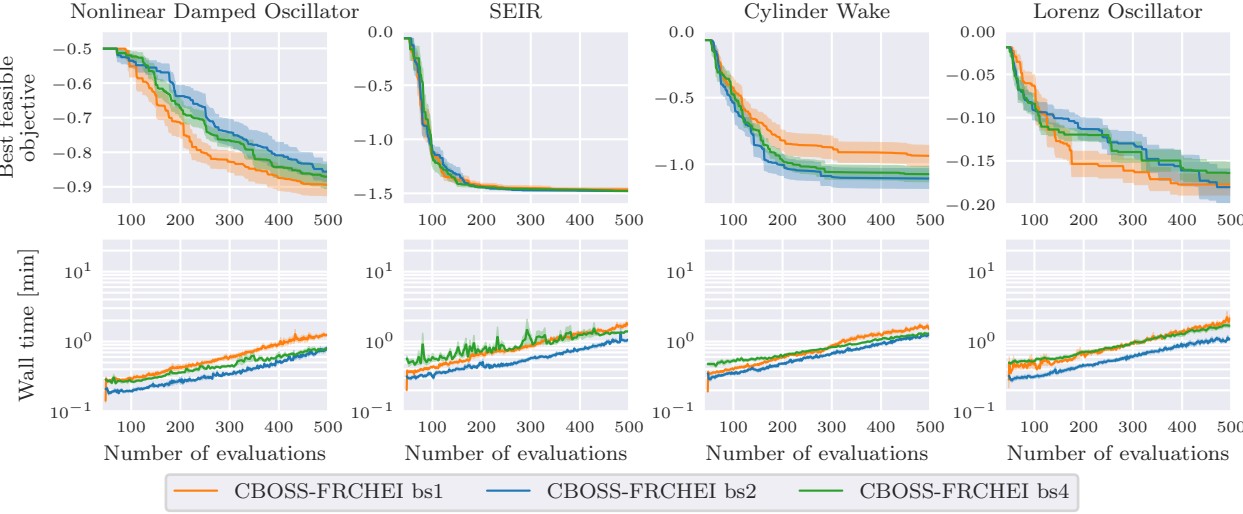

Figure 15: Optimization results for `CBOSS-FRCHEI` method for different batch sizes of 1, 2 and 4.

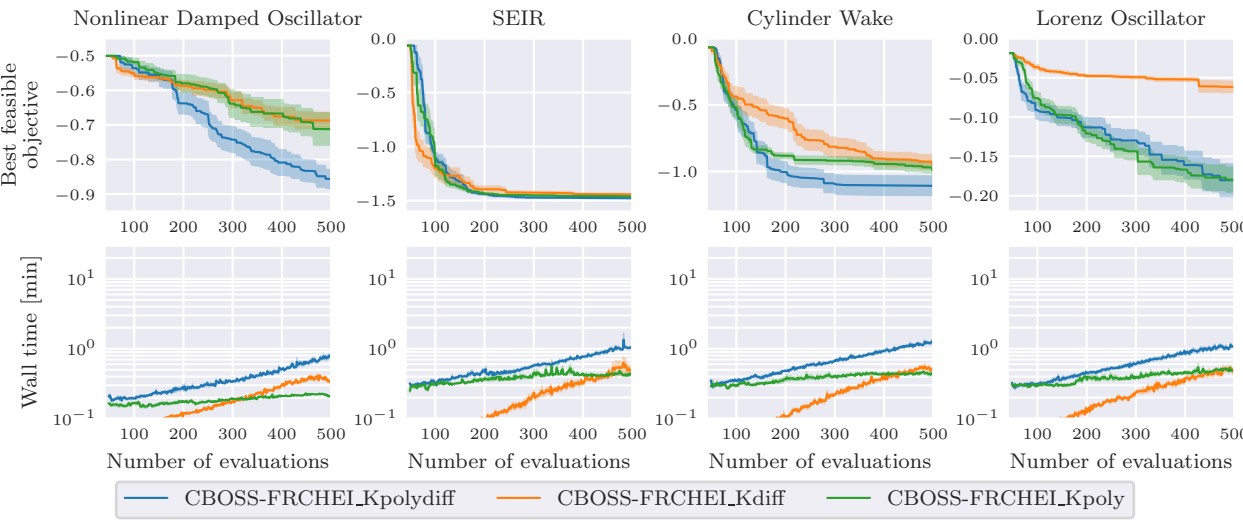

Figure 16: Optimization results for `CBOSS-FRCHEI` method with different kernel functions: Kpoly is the polynomial kernel, Kdiff the discrete diffusion kernel and Kpolydiff the combination of the two.

