# OpenReview forum: "Discovering Model Structure of Dynamical Systems with Combinatorial Bayesian Optimization"
_TMLR — Accepted by TMLR_

### Review · Reviewer_FHKn · 2023-11-08

**Summary Of Contributions:**

A version of Bayesian optimization (BO) is suggested, where inequality constraints (this is somewhat easy) and failure of measurements (this is much harder) are taken into account when applying BO on discrete spaces. The BO approach build on state of the art changes to Gaussian process models, combined with changes to the acquisition function. The approaches is tested on two non-trivial applications (differential equations learning and model selection for car dynamics).

**Audience:**

Yes

**Claims And Evidence:**

Yes

**Requested Changes:**

Try to rewrite the text to make it more concise.

Minor:
 - Between (7) and (8): Var -> \operatorname{Var}
 - I do not think that p(theta)=1 is the correct Jeffrey's prior for all kinds of variables.

**Strengths And Weaknesses:**

Strengths:
 - The paper uses a long list of state of the art methods in machine learning. The usage of these methods is sound and well-motivated, and elevates above similar papers that use rather standard methods.
 - The state of the art methods used for comparison are strong methods (perhaps except SA).
 - The examples for evaluation are strong examples. Repacing SINDy for learning differential equations from data is a major strong point of this paper.

Weaknesses:
 - Written rather wordy. Not critically, but the text might be shortened significantly. In particular, for a journal that strives for short novelty papers, this paper seems much more as a review of the state of the art in several domains and doing a combination of several known techniques. Case in point (this is everywhere else, too) is the last paragraph of page 14, which I could rewrite as "Figures 7a and 7b show how our method consistently outperforms others in optimizing models across multiple runs and benchmark problems, with low variance – crucial for expensive one-time optimization tasks. Even the pure exploratory policy underperforms. A balanced method like CBO-FRCHEI strikes the right trade-off between exploration and exploitation, effectively navigating model structures and avoiding local minima.". This is half as long as the original text.
 - Modeling parameters and model structure both in the discrete space X seems rather restrictive to me. First, because a model is only allowed to have discrete parameters. Second, because the model structure does not change the model parameters, which is rather typical in my experience. (Since I see no reasons why the methods could not be combined with classical methods on continuous spaces, a short comment in this direction might easily fix this weakness.)
 - While there are ablation studies, there could be more. Many choices are done in this paper. However, the paper is rather long anyway, and the choices in the paper look rather natural, I see this as a minor point.

Questions:
 - Why is h a binary equality constraint function? This might make optimization harder. Of course, there are scenarios where this is the case. Perhaps you can comment on which scenarios you have in mind and why/how you are different from the literature. You approach reminds me of some papers I read about safe active learning, perhaps you can comment on differences and connections.
 - EI as acquisition function is no longer state of the art. Can alternative methods be used instead? (This is a rare occasion, where the choise of the method is not state of the art.)
 - How do you construct priors for hyperparameters specifically?

In general, I think the content of the paper is publishable. The methods are not completely novel, but a non-trivial and strong combination of rather new methods in a non-obvious manner. The language could be improved.

---

> ### Author Response · Authors · 2024-01-25
> **Author Response (Part 1/2)**
>
> > - Modeling parameters and model structure both in the discrete space X seems rather restrictive to me. First, because a model is only allowed to have discrete parameters. Second, because the model structure does not change the model parameters, which is rather typical in my experience. (Since I see no reasons why the methods could not be combined with classical methods on continuous spaces, a short comment in this direction might easily fix this weakness.)
>
> We decide to separate the task of model structure selection and parameter estimation into a bilevel optimization problem. This allows for both continuous and discrete parameters. In this paper we focus on discrete parameters since the model structures in our problems are naturally defined using categorical decision variables. In this sense our method is not restrictive to only discrete spaces and for the continuous parameters any appropriate optimization algorithms can be chosen. For the synthetic examples we choose a least square algorithm but we could also use Bayesian optimization. Importantly, a different model structure will, generally, lead to different model parameters and our method allows for this.
> The separation also allows that different model structures can have different continuous parameters.
>
> We agree that it is possible to extend our model structure optimization method to handle other than binary or categorical variables. These mixed spaces are however more challenging. First, one must consider kernels that can capture the interaction between discrete and continuous variables. Second, optimizing the acquisition function over mixed spaces would requires expensive interleaved optimization schemes.
> Another choice is to use probabilistic reparameterization (PR) of the discrete variables. In our result we show that the proposed method is competitve with PR or outperforms it while having a much lower computational burden.
>
> > - Why is h a binary equality constraint function? This might make optimization harder. Of course, there are scenarios where this is the case. Perhaps you can comment on which scenarios you have in mind and why/how you are different from the literature. You approach reminds me of some papers I read about safe active learning, perhaps you can comment on differences and connections.
>
> The binary equality constraint is used in this paper as a formal model to handle failures and crashes. This does makes the optimization harder. However, we need such a constraint to model the problems we consider in this paper, where these constraint are present.
> For the model structure identification task we consider in this paper many configurations resulted in model structures that led to unstable dynamical systems. These configurations were labeled with $l=h(\boldsymbol x)=1$. In essence, we have to solve a classification problem (modeled as the equality constraint) and an optimization problem.
>
> We are using a GP classifier to learn the failure regions in order to account for this constraint and multiple the acquisition with the probability of constraint satisfaction (see Section 4.3).
> There are also other cases, when this binary equality constraint function can be helpful when searching for the best model configuration. It might happen that the parameter estimation phase does not converge or lead to unsatisfactory or non-physical values. Even if it would be possible to assess the objective and constraint function values at the configuration $\boldsymbol x$, it is reasonable to avoid selecting these model structures. Thus, there is an option for the user of the optimizer to use this as an artifact to make the search for better structures more efficiently and guarantee that the final model structure is valid.

---

> ### Author Response · Authors · 2024-01-25
> **Author Response (Part 2/2)**
>
> > - EI as acquisition function is no longer state of the art. Can alternative methods be used instead? (This is a rare occasion, where the choise of the method is not state of the art.)
>
> Although there are plenty of other acquisition functions in the literature, EI has been chosen since it provides an elegant and simple way to incorporate constraints and handle failures. EI enjoys both theoretical guarantees as well as empirical evidence that it works well with constraints and crashes. See Gardner et al (2014) and Chakrabarty (2021).
> The main benefit in our work is that EI, even with constraints and crashes, is given in closed-form and can be computed efficiently. Further, replacing GPs by TPs also retains the closed-form solution. While we developed the our method we have seen that if we are not able to optimize the acquisition function well the method will yield much worse results.
>
> In the presented work we have to find the optimum of the acquisition function in a combinatorial space with trillions of possible combinations, with no gradients due to the discrete space. This requires to run simulated annealing for many steps. To avoid running in to computational issues where the optimization of the acquisition function takes much longer then the simulation of a new model structure it is crucial that the evaluation of the acquisition function is fast.
>
> > - How do you construct priors for hyperparameters specifically?
>
> Such a discussion was indeed missing from the paper. We have added a dedicated `Hyperparameters` section to the appendix `A.2` where we list and discuss the hyperparameters and its hyperpriors. We also include plots and brief motivations for the chosen values, as well as plots to guide the reader through the choices made.
>
> > Try to rewrite the text to make it more concise.
>
> We have rewritten a large portion of the text to make it more concise. We have reduced the size of an entire page while retaining the same information.
>
> > - Between (7) and (8): Var -> \operatorname{Var}
>
> This has been fixed.
>
> > I do not think that p(theta)=1 is the correct Jeffrey's prior for all kinds of variables.
>
> We agree with the reviewer that the term Jeffrey's prior is misleading. We have replaced the term _Jeffrey's prior_ by _uninformative prior_.

---

### Review · Reviewer_doWD · 2023-12-05

**Summary Of Contributions:**

The paper introduces a novel approach to model structure selection, framing it as a black-box constrained combinatorial optimization problem. Crucially, they use a mix of *categorical decision variables* and *continuous variables* to *select model* templates as well as its *free parameters*. Additionally, the proposed method addresses can also handle inequality and failures constraints, which the authors referred by "crash constraints" in the paper.

The proposed algorithm, CBO-FRCHEI, is designed to efficiently solve complex model structure selection problems. The key components of the proposed methodology are:
* Using a t-process regression instead of a regular Gaussian Process.
* Developing a custom kernel that combines two previously proposed kernels for discrete spaces: the discrete diffusion kernel and the truncated polynomial kernel.
* An acquisition function called Failure-Robust Constraint Hierarchical Expected Improvement that uses Expected Improvement using the t-process regression and multiplies it with the probability of success (not experiencing a failure or crash) and the probability of feasibility (satisfying all the constraints).

Empirical evidence presented in the paper demonstrates that CBO-FRCHEI outperforms existing state-of-the-art methods in benchmark problems related to equation discovery. The paper extends its contributions to practical applications by showcasing the construction of a digital twin for a driving simulator using CBO-FRCHEI.

**Audience:**

Yes

**Broader Impact Concerns:**

No broader impact concerns.

**Claims And Evidence:**

Yes

**Requested Changes:**

- I apologize if I overlooked this, but it seems that the proposed acquisition function has hyperparameters βsucc and βfeas. I couldn't find a discussion on how these parameters were determined. Could you kindly provide this information and offer recommendations to assist readers in understanding their impact?
- Please consider revising the text to ensure that the novelty of the proposed acquisition function is not overly emphasized.
- For completeness, consider discussing the following related work:
  1. Gardner, J.R., Guo, C., Weinberger, K.Q., Garnett, R., & Grosse, R. (2017). Discovering and Exploiting Additive Structure for Bayesian Optimization. International Conference on Artificial Intelligence and Statistics (AISTATS 2017).
  2. Duvenaud, David, Lloyd, James Robert, Grosse, Roger, Tenenbaum, Joshua B., & Ghahramani, Zoubin (2013). International Conference on Machine Learning, 2013.
  3. Malkomes, G., Schaff, C., & Garnett, R. (2016). Bayesian Optimization for Automated Model Selection. In Advances in Neural Information Processing Systems. (NeurIPS 2016).
  4. Raissi, M., & Karniadakis, G.E. (2018). Hidden physics models: Machine learning of nonlinear partial differential equations. Journal of Computational Physics, Volume 357, 15 March 2018, Pages 125-141.

**Strengths And Weaknesses:**

**Strength**. This paper is quite well-written, and the proposed method is well motivated. The algorithmic decisions are properly contextualized and appear quite intuitive. I like the idea of the T-Process as well as the use of  categorical decision varibles for selecting strctures. The normalization of the P_{succ} and P_{feas} for n/N is also interesting. In the experiment section, I especially appreciate the inclusion of both sets of experiments, which showcase the discrete model search under investigation and its applicability.

**Weakness**. I don't think the proposed methodology is particularly novel. The main surrogate model, T-Process, is from a previous work; and the combination of P_{succ}, P_{feas} and any version of EI is fairly common.
The main strenght of this paper is how well they have placed all the components together to solve an interesting problem.

I am also surprise with the high number of binary and categorical variables used here. The authors used simulated annealing (SA) to optimize the acqusition function. I'm wondering if the authors have consider other methodologies for this optimization or could provide any insights into why the SA proposed by Dadkhahi (2022) is a good choice here.

---

> ### Author Response · Authors · 2024-01-25
> **Author Response**
>
> > The authors used simulated annealing (SA) to optimize the acqusition function. I'm wondering if the authors have consider other methodologies for this optimization or could provide any insights into why the SA proposed by Dadkhahi (2022) is a good choice here.
>
> SA has been the standard acquisition function optimizer in combinatorial BO and we have not considered alternatives. The benefits of using SA as an acquisition function optimization are: (i) it is a computationally cheap, (ii) strong empirical performance, (iii) convergence guarantees, (vi) very few hyperparameters, and (v) local exploration. All these features are crucial to select informative queries in combinatorial spaces with trillions of possible combinations.
> The version proposed by Dadkhahi (2022) is an direct extension to the traditional SA that allows optimizing over categorical spaces.
>
> Alternative acquisition function optimizer for discrete spaces are a potential direction for future research and could improve existing combinatorial BO methods including the one we present in this paper.
>
> > I apologize if I overlooked this, but it seems that the proposed acquisition function has hyperparameters βsucc and βfeas. I couldn't find a discussion on how these parameters were determined. Could you kindly provide this information and offer recommendations to assist readers in understanding their impact?
>
> Such a discussion was indeed missing from the paper. We have added a dedicated appendix section `A.2 Hyperparameters`, including plots and brief discussion to build some intuition of how these hyperparameters influence the acquisition function.
>
> > Please consider revising the text to ensure that the novelty of the proposed acquisition function is not overly emphasized.
>
> We have revised the text and modified the abstract and the contribution list as requested.
>
> > For completeness, consider discussing the following related work:
> > 1.  Gardner, J.R., Guo, C., Weinberger, K.Q., Garnett, R., & Grosse, R. (2017). Discovering and Exploiting Additive Structure for Bayesian Optimization. International Conference on Artificial Intelligence and Statistics (AISTATS 2017).
> > 2. Duvenaud, David, Lloyd, James Robert, Grosse, Roger, Tenenbaum, Joshua B., & Ghahramani, Zoubin (2013). International Conference on Machine Learning, 2013.
> > 3. Malkomes, G., Schaff, C., & Garnett, R. (2016). Bayesian Optimization for Automated Model Selection. In Advances in Neural Information Processing Systems. (NeurIPS 2016).
> > 4.  Raissi, M., & Karniadakis, G.E. (2018). Hidden physics models: Machine learning of nonlinear partial differential equations. Journal of Computational Physics, Volume 357, 15 March 2018, Pages 125-141.
>
> We extended the `Model Strucutre Selection` paragraph in the `Related Work` section to include the references 1-3.
>
> The forth reference proposes a method for parameter estimation of known model structures, here partial differential equations. While the paper is certainly interesting and very relevant for system identification it doesn't address the problem we are tackling in our paper: finding the model structure.
> We have therefore decided to not include this reference.

---

### Review · Reviewer_1qh9 · 2024-01-19

**Summary Of Contributions:**

In this paper, the authors' propose a combinatorial Bayesian optimization algorithm with both inequality and equality constraints. The authors' motivating application domain is the construction of models for dynamical systems. To achieve this, they assume access to a set of model "templates" that can be combined and then parameter fit. The black-box objective optimized using Bayesian optimization is a measure of goodness-of-fit of the resulting model after both model template selection and parameter estimation, under constraints like resource requirements (inequality) and lack of model failure (equality).

The authors algorithm consists of:
- A hierarchical Gaussian process model with an inverse Gamma prior over the outputscale (I think -- see my comment below under "minor" about the heavy use of \sigma^2) which leads to multivariate Student's t-distributed predictive distributions. The authors use a linear mixing of a product of the kernels of Deshwal et al., 2021 and Baptista & Poloczek 2018, and an additive kernel of these same two kernels.

- An acquisition function that is a cEI-like combination of a "hierarchical EI" acquisition function (eqns. 21 and 22) with the product of feasibility probabilities under the constraint surrogates.

**Audience:**

Yes

**Broader Impact Concerns:**

I don't see any significant concerns here.

**Claims And Evidence:**

No

**Requested Changes:**

Ablation:
Maybe I am missing something here, but it seems like the authors' method using a GP surrogate model (FRCEI) has comparable or (possibly statistically significantly!) better performance on three out of four tasks in your ablation study (Figure 11)? The authors even mention this ("FRCEI provides similar performance and sometimes slightly outperforms FRCHEI"), but the authors conclude that t processes are nevertheless a "more robust and flexible probabilistic model." Is there actually compelling evidence of this? Certainly not in Figure 11. I'd really like to see the paper do a better job of actually supporting the authors' own modelling choices.

Related work and baselines:
The authors' related works and literature survey is reasonably good for combinatorial Bayesian optimization methods. The only missing relatively recent work that I'm aware of is BODi (Deshwal et al., 2023). BODi has quite good performance in their own results; however, it does not directly handle constraints I think.

I'm not particularly enthusiastic about the authors' claims that other combinatorial bayesopt methods can't be compared to because they don't mention or handle constraints in their own paper. For many of these methods, the key innovation had nothing to do with acquisition, which is often where constraints are handled. For example, BODi uses EI in a straightforward manner -- there's nothing really preventing the use of cEI e.g. as in Gardner et al., 2014 or constrained Thompson sampling as in Eriksson et al., 2019.

With that said, PR is a reasonably competitive baseline in this area, and the authors' method does seem to perform quite favorably on a few of the tasks they consider.

Minor:
- The symbol \sigma^2 is extremely overloaded. On page 6 alone it is used variously to refer to: the likelihood noise for the objective function (\sigma^2_y), the likelihood noise for the constraint functions (\sigma^2_c), an outputscale (\sigma^2), as what I think is a typo again for the likelihood noise (\sigma^2 -- should be \sigma^2_y in equation 5?), a scale parameter for an inverse Gamma distribution (\sigma^2_m). The likelihood noise is then I think dropped in equation (9)? Can the authors take a quick pass through and clean some of that up, or verify correctness here?

- Wan et al., 2021 technically called their method "Casmopolitan" (note the a).

- Algorithm 2 is maybe superfluous? An algorithm box that contains "train the surrogate model, optimize the acquisition function, evaluate objectives and constraints, repeat" is essentially just a description of Bayesian optimization, which already exists well enough in the text.

**Strengths And Weaknesses:**

Overall, the authors' method seems to achieve good performance, and the authors' experimental evaluation is reasonably thorough. I think the modelling choices are mostly fine -- t-Processes are good models if a bit annoying to do hyperparameter learning with sometimes, and the authors' acquisition function looks reasonable if standard. In particular, I haven't seen very many Bayesian optimization papers using t-Processes, thinking of how acquisition functions might change under t-Process models, or comparing them to standard GP based Bayesian optimization. Therefore, the comparison alone is pretty interesting to me.

Beyond my interest in the methodology, the authors' consider a host of fairly interesting real world optimization problems that seem quite challenging.

For weaknesses, I have a few comments mostly centering around some of the experiments, and the ablation studies in particular are very confusing to me, and indeed seem to contradict some of the authors' messaging about t-Processes in their methods section (see below).

---

> ### Author Response · Authors · 2024-01-25
> **Author Response**
>
> > Maybe I am missing something here, but it seems like the authors' method using a GP surrogate model (FRCEI) has comparable or (possibly statistically significantly!) better performance on three out of four tasks in your ablation study (Figure 11)? The authors even mention this ("FRCEI provides similar performance and sometimes slightly outperforms FRCHEI"), but the authors conclude that t processes are nevertheless a "more robust and flexible probabilistic model." Is there actually compelling evidence of this? Certainly not in Figure 11. I'd really like to see the paper do a better job of actually supporting the authors' own modelling choices.
>
> We thank the reviewer for the feedback. We have reformulated our claims based on the empirical results. Indeed, TPs do not clearly outperform GPs in our test cases. We have adjusted the paper to include both models, GPs and TPs, as possibilities for model structure selection in system identification.
> Even though the TP model did not clearly outperform the formulation with GPs in the benchmark problems we investigate, we still prefer the TP formulation because it is a more general model that can be more robust against outliers, and it does not increase the implementation complexity or computation time. For the paper we leave the choice of model to the user and provide now provide empirical results for both in Section 5.1 and briefly discussed their performances. The potential implications of using TP on the results are also discussed in the appendix section.
> Throughout the text we now mention that the there are two options within our framework.
>
>
> > Related work and baselines: The authors' related works and literature survey is reasonably good for combinatorial Bayesian optimization methods. The only missing relatively recent work that I'm aware of is BODi (Deshwal et al., 2023). BODi has quite good performance in their own results; however, it does not directly handle constraints I think.
>
> Thanks for pointing out this paper. We have added it to the related work section.
>
> > I'm not particularly enthusiastic about the authors' claims that other combinatorial bayesopt methods can't be compared to because they don't mention or handle constraints in their own paper. For many of these methods, the key innovation had nothing to do with acquisition, which is often where constraints are handled. For example, BODi uses EI in a straightforward manner -- there's nothing really preventing the use of cEI e.g. as in Gardner et al., 2014 or constrained Thompson sampling as in Eriksson et al., 2019.
> > With that said, PR is a reasonably competitive baseline in this area, and the authors' method does seem to perform quite favorably on a few of the tasks they consider.
>
> The main reason not to compare to more algorithms is the very large computational burden. This makes them infeasible for our use cases and computational resources. The fact that they do not support constraint is a secondary concern. We'd have to reimplement their methods with constraints, and instead chose PR as a competitive baseline. We have reformulated our claims in the text to clarify the reasoning behind not including other methods in our comparison.
>
>
> > - The symbol \sigma^2 is extremely overloaded. On page 6 alone it is used variously to refer to: the likelihood noise for the objective function (\sigma^2_y), the likelihood noise for the constraint functions (\sigma^2_c), an outputscale (\sigma^2), as what I think is a typo again for the likelihood noise (\sigma^2 -- should be \sigma^2_y in equation 5?), a scale parameter for an inverse Gamma distribution (\sigma^2_m). The likelihood noise is then I think dropped in equation (9)? Can the authors take a quick pass through and clean some of that up, or verify correctness here?
> > - Wan et al., 2021 technically called their method "Casmopolitan" (note the a).
>
> This has been fixed in the new version of the paper.
>
> > - Algorithm 2 is maybe superfluous? An algorithm box that contains "train the surrogate model, optimize the acquisition function, evaluate objectives and constraints, repeat" is essentially just a description of Bayesian optimization, which already exists well enough in the text.
>
> We include Algorithm 2 for completeness and to summarize the information in the text. While for a BO expert it might be slightly superfluous we hope its inclusion  facilitates the understanding of the application of the method in the context of model structure selection in system identification.
> We therefore decided to keep it in the main text.

---

### Author Response · Authors · 2024-01-25
**General comments**

We thank the reviewers for the valuable and constructive feedback and comments. We have uploaded a new version of the PDF addressing the issues raised in this review.

Detailed answers to all questions can be found in the responses. Here is an overview of the main changes in the new version. We have
- renamed the method to CBOSS
- rewrote the text to make it more concise
- moved the results of our method with the GP regression surrogate (CBOSS-FRCEI) to the main paper
- added a new appendix section A.2 Hyperparameters

---

### Decision · Action_Editor_3L3L · 2024-03-01

**Recommendation:** Accept as is

**Comment:**

This manuscript considers the problem of model structure optimization for modeling dynamical systems, a challenging and important problem. The authors propose to apply techniques from Bayesian optimization to this problem, a natural but not at all straightforward idea. The authors outline a well-motivated and feature-rich framework for applying Bayesian optimization to the problem of interest, and validate this approach through a series of well-designed experiments.

The reviewers agree that the work presented in this manuscript is high quality and of interest to the TMLR audience. They are universal in their recommendation of acceptance.

Throughout the review and discussion period, the reviewers provided feedback and suggestions that the authors have faithfully incorporated into the manuscript. I believe these changes have considerably strengthened the work. I commend both authors and reviewers for their engagement throughout this process.

I believe the manuscript is both suitable for and ready for publication.

**Audience:**

There is no question that the material in this paper would be of interest to a subset of TMLR's audience as it concerns the fundamental and widely studied framework of Bayesian optimization. Further, the chosen application – model selection for dynamical systems – is itself an important problem that would be of relevance to a separate, not completely overlapping, subset of TMLR's audience.

**Claims And Evidence:**

The reviewers universally agree that the claims made in the submission are supported by accurate, convincing, and clear evidence, both in the form of clear exposition and well-designed empirical studies.